# Dissolved organic matter and its optical characteristics in the Laptev and East Siberian seas: Spatial distribution and inter-annual variability (2003-2011)

Svetlana P. Pugach[1,2], Irina I. Pipko[1,2], Natalia E. Shakhova[2,3], Evgeny A. Shirshin[4], Irina V. Perminova[5], Örjan Gustafsson[6,7], Valery G. Bondur[8], Alexey S. Ruban[2], Igor P. Semiletov[1,2,3]

[1]V.I. Il'ichev Pacific Oceanological Institute, Russian Academy of Sciences, Vladivostok, 690041, Russia
[2]National Research Tomsk Polytechnic University, Tomsk, 634050, Russia
[3]International Arctic Research Center, University of Alaska Fairbanks, Fairbanks, AK 99775, USA
[4]Department of Physics, Lomonosov Moscow State University, Moscow, 119991, Russia
[5]Department of Chemistry, Lomonosov Moscow State University, Moscow, 119991, Russia
[6]Department of Environmental Science and Analytical Chemistry, Stockholm University, Stockholm, 10691, Sweden
[7]Bolin Centre for Climate Research, Stockholm University, Stockholm, 10691, Sweden
[8]Aerocosmos Research Institute of Airspace Monitoring, Moscow, 105064, Russia

*Correspondence to* Svetlana P. Pugach (pugach@poi.dvo.ru)

**Abstract.** The East Siberian Arctic Shelf (ESAS) is the broadest and shallowest continental shelf in the World Ocean. It is characterized by both the highest rate of coastal erosion in the world and a large riverine input of terrigenous dissolved organic matter (DOM). DOM plays a significant role in marine aquatic ecosystems. The chromophoric fraction of DOM (CDOM) directly affects the quantity and spectral quality of available light, thereby impacting both primary production and ultraviolet (UV) exposure in aquatic ecosystems.

A multi-year study of CDOM absorption, fluorescence, and spectral characteristics was carried out over the vast ESAS in the summer-fall season. The paper describes observations accomplished at 286 stations and 1766 in situ high-resolution optical measurements distributed along the nearshore zone. Spatial and inter-annual CDOM dynamics over the ESAS were investigated, and driving factors were identified. It was shown that the atmospheric circulation regime is the dominant factor controlling CDOM distribution on the ESAS.

This paper explores the possibility of using CDOM and its spectral parameters to identify the different biogeochemical regimes in the surveyed area. The analysis of CDOM spectral characteristics showed that the major part of the Laptev and East Siberian seas shelf is influenced by terrigenous DOM carried in riverine discharge. Western and eastern provinces of the ESAS with distinctly different DOM optical properties were also identified; a transition between the two provinces at around 165-170° E, consistent also with hydrological and hydrochemical data.

In the western ESAS, a region of substantial river impact, the content of aromatic carbon within DOM remains almost constant. In the eastern ESAS, a gradual decrease in aromaticity percentage was observed indicating contribution of Pacific-origin waters, where allochthonous DOM with predominantly aliphatic character and much smaller absorption capacity predominates. In addition, we found a stable tendency towards reduced concentrations of CDOM and dissolved lignin, and increase of spectral slope and slope ratio values eastward from the Lena River delta; the Lena is the main supplier of DOM to the Eastern Arctic shelf.

The strong positive correlation ($r = 0.97$) between dissolved organic carbon (DOC) and CDOM values in the surface shelf waters influenced by terrigenous discharge indicates that it is feasible to estimate DOC content from CDOM fluorescence assessed in situ using a WETStar fluorometer. This approach is reliable over the salinity range of 3 to 24.5. The fact that there is little difference between predicted and observed parameters indicates that the approach is justified. The direct estimation of DOM optical characteristics in the surface ESAS waters provided by this multi-year study will also be useful for validating and calibrating remote sensing data.

## 1 Introduction

Current climate change is particularly evident and amplified in the high latitudes of the Northern Hemisphere. The system change is characterized by an increase in the average annual temperature and atmospheric circulation intensity, a reduction in sea ice coverage and thickness, and accelerated degradation of permafrost, as well as an increase in coastal and bottom erosion and river flows (IPCC, 2013). Unlike other oceans, the Arctic Ocean is completely surrounded by permafrost. The Arctic region contains an abundance of organic carbon (OC) buried inland and within the sedimentary basin of the Arctic Ocean, which might become a part of the current marine biogeochemical cycle due to thawing of on-land and sub-sea permafrost, increased coastal and bottom erosion, accelerated river discharge, and soil-based carbon losses (Günther et al., 2013; Shakhova et al., 2009, 2017; Semiletov, 1999; Vonk and Gustafsson, 2013; Vonk et al., 2012, 2014). The recent studies accomplished within the framework of the International Siberian Shelf Study project (Semiletov and Gustafsson, 2009; Semiletov et al., 2011; Tesi et al., 2014; Bröder et al., 2016; Charkin et al., 2011) demonstrate that coastal erosion is the main source of particulate OC (POC) to the East Siberian Arctic Shelf (ESAS), the broadest and shallowest shelf in the World Ocean. Oxidation of POC exported from thawing coastal (and bottom) permafrost, and freshening due to growing Siberian river runoff from extensive permafrost-underlain watersheds, play a major role in the severe ESAS acidification that has been reported (Semiletov et al., 2016). At the same time, river discharge to shelf waters supplies terrestrial carbon in the form of dissolved OC (DOC) and waters enriched by carbon dioxide ($CO_2$) (Alling et al., 2010; Amon et al., 2012; Anderson et al., 2009, 2011; Semiletov et al., 2011, 2012; Pipko et al., 2010).

Annually, the Arctic rivers transport 25-36 Tg of DOC to the Arctic Ocean, which is ~10 % of the global riverine DOC discharge (Raymond et al., 2007). Siberian rivers are high in DOC, with a mean concentration of more than 500 µM (Gordeev et al., 1996; McClelland et al., 2012; Amon et al., 2012). These concentrations are an order of magnitude higher than in the inflowing Atlantic (60 µM) and Pacific (70 µM) waters, but the volume flux of the two oceans is about 60 times larger than that of continental runoff (Anderson and Amon, 2015). Furthermore, Arctic and subarctic regions contain approximately 50 % of the global terrestrial OC in their frozen soils (Hugelius et al., 2011). Warming and intensification of the hydrologic cycle is leading to an increased rate of water and dissolved organic matter (DOM) discharge from the Siberian rivers (Savelieva et al., 2000; Semiletov et al., 2000; Stein and Macdonald, 2004).

The absorbance and fluorescence properties of the chromophoric fraction of DOM (CDOM) are "optical markers", comparable to traditional biomarkers used in geochemistry (e.g., lignin) (Stedmon and Nelson, 2015). According to Coble (2007), up to 70 % of DOM in shelf waters is represented by CDOM, which is critical in a number of biochemical and photochemical processes and which defines the optical properties of natural waters, thus affecting the depth of the photic layer (Granskog et al., 2007; Hill, 2008). Quantitative descriptions of the dynamics and variability of CDOM optical properties are often required, particularly in coastal waters, in order to accurately predict light penetration and primary production (Matsuoka et al., 2007). It should be noted that reliable estimation of coastal water optical characteristics is crucial for validating and calibrating remote sensing data processing results (Vantrepotte et al., 2012; Bondur and Vorobev, 2015). High CDOM values are typical for the eastern Laptev Sea; in the western East Siberian Sea they result from large discharge of the Lena River which is characterized by high CDOM concentrations (Semiletov et al., 2013; Pugach and Pipko, 2013). Most likely, the color that indicates the presence of CDOM in this water, when seen from space, is due to the presence of chlorophyll (Heim et al., 2014). This may explain the recently proposed increase in the net primary production rates reported for these ESAS areas based on satellite data interpretation (Arrigo and van Dijken, 2011).

Empirical relationships between the optical properties of DOM and DOC have already been the subject of investigation in the Arctic Ocean (Stedmon et al., 2011; Guéguen et al., 2005, 2007; Fichot and Benner, 2011; Gonçalves-Araujo et al.,

2015; Kaiser et al., 2017). The present study synthesizes the authors' multi-year observations on the remote ESAS with the focus on exploring the extent and dynamics of riverine DOC, using CDOM as a proxy.

This paper aims (1) to study the spatial and inter-annual dynamics of DOM optical characteristics in shelf waters of the Eastern Arctic seas on the basis of multi-year summertime (August –September) expedition data (2003, 2004, 2005, 2008, 2011); (2) to examine the relationship between CDOM fluorescence and DOC in order to validate a useful method for accurately predicting DOC concentration from CDOM properties in the ESAS; and (3) to demonstrate the feasibility, using DOM optical characteristics, of determining the terrigenous DOM distribution and identifying different biogeochemical provinces in the shelf water.

## 2 Study area

This study is focused on the Laptev and East Siberian seas, where the influence of river discharge and the biogeochemical signal of permafrost degradation are the most prominent (Alling et al., 2010; Nicolsky and Shakhova, 2010; Sánchez-Garcia et al., 2011; Shakhova et al., 2015, 2017; Vonk et al., 2012). The shelf area of these seas covers 40% of the Arctic shelf and 20% of the entire Arctic Ocean area (Stein and Macdonald, 2004). Annually, 767 km$^3$ of fresh water flows into the Laptev Sea with discharges of the Lena, Khatanga, Anabar, Olenek, and Yana rivers; this amounts to over 30 % of the total river discharge into all Arctic seas in Russia. The Lena River, which is the largest river in the ESAS region, accounts for more than 70 % of the total discharge to the Laptev Sea (Semiletov et al., 2000; Cooper et al., 2008). Over 85 % of its discharge flows into the sea through channels of the eastern Lena delta. Therefore, the hydrological and biogeochemical regime of the Laptev Sea, and its southeast part in particular, is mainly controlled by discharge fluctuations of the Lena River (Pipko et al., 2010; Semiletov et al., 2011, 2012; Bröder et al., 2016). Moreover, it has been found that in the past the Lena River played a dominant role in sediment discharge, flushing out soil OM from its vast watershed (Tesi et al., 2016); a significant fraction of "fresh" terrestrial OM contributes to the DOM pool (Vonk et al., 2013; Karlsson et al., 2016).

The East Siberian Sea is one of the least-studied seas; this lack of attention is mainly attributed to severe ice-bound conditions. The Kolyma and Indigirka are the main rivers for which the East Siberian Sea serves as a receiving basin; however, their total average annual discharge does not exceed 180 km$^3$. According to Nikiforov and Shpaikher (1980) a major part of the Lena River fresh water plume usually transits the East Siberian Sea eastward within the coastal zone. Depending on hydrometeorological conditions, this plume reaches 165° E (typical under anticyclonic circulation conditions), and as far as Long Strait at 180° E (typical under cyclonic circulation conditions) (Semiletov et al., 2005).

## 3 Materials and methods

### 3.1 Field work

This paper presents data from research expeditions conducted during the summer-fall seasons of 2003 and 2004 (HV *Ivan Kireev*), 2005 (MV *Auga*), 2008 (HV *Yakob Smirnitsky*), and 2011 (RV *Academician M.A. Lavrentyev*) on ice-free regions of the Laptev and East Siberian seas (Fig. 1).

The study was carried out using a hydrological conductivity-temperature-depth (CTD) SBE 19plus SEACAT Profiler probe (www.seabird.com) with CDOM sensor (WETStar DOM Fluorometer, www.wetlabs.com). This enabled in situ synchronous high-precision measurements of vertical temperature distribution, conductivity, and CDOM.

In 2003, 2004, 2005, and 2011 the measurements were carried out at 286 oceanological stations and in 2008 along the ship's track using a seawater intake (SWI) system. Water was pumped from 4 m depth at 30 L min$^{-1}$ through stainless steel and silicon tubes into an on-deck 300 L barrel, and then through a distribution network, from which samples were collected. Temperature, salinity/conductivity, and CDOM in the surface water were measured continuously using sensors installed on

the CTD Seabird 19+ deployed in the barrel; measurements were averaged every 10 min. Table 1 shows detailed information on cruises, including dates, number of stations, and parameters measured.

## 3.2 Methods for determining dissolved organic carbon

In 2004, DOC concentration was measured in an International Arctic Research Center laboratory at the University of Alaska (Fairbanks, Alaska, USA), in 2008 on board the RV *Yakob Smirnitsky* (Table 1). All samples were analyzed using the Shimadzu TOC-V$_{CPH}$ system (Alling et al., 2010). The 1-3 L seawater samples were vacuum-filtered onboard with 25 mm diameter pre-combusted borosilicate glass fiber filters (Whatman GF/F, nominally 0.7 μm cut-off) within an all-glass filtration system. The samples were stored in 60 mL Nalgene high-density polyethylene bottles. DOC was analyzed via high-temperature catalytic oxidation. Inorganic carbon was removed by acidifying the samples to pH 2 with 2 M HCl and sparging for 8 min prior to analysis. All procedures for calibration and data analysis followed Sharp et al. (1995). Certified Reference Materials (CRMs, from the University of Miami) of low carbon content (1-2 μM C) and deep-sea reference water (41-44 μM C) were run prior to each analysis batch (Alling et al., 2010).

## 3.3 Absorbance and fluorescence analysis

Absorbance of CDOM was measured using a UNICO 2804 spectrophotometer with a 1 cm quartz cuvette over the spectral range from 200 to 600 nm at 1 nm intervals (Table 1). Milli-Q (Millipore) water was used as the reference for all samples. Water samples underwent filtration through acid-washed Whatman glass fiber filters (GF/F, nominal pore size 0.7 μm).

The absorption coefficient ($a_\lambda$, m$^{-1}$) was calculated as follows:

$$a_\lambda = 2.303 A_\lambda / L, \tag{1}$$

where $A_\lambda$ is optical density at wavelength $\lambda$, and L is the cell pathlength in meters.

The absorption coefficients at 350 nm ($a_{350}$) were chosen to quantify the concentrations of CDOM because of their correlations to DOC, and to permit comparison with other results (Spencer et al., 2009; Stedmon et al., 2011; Walker et al., 2013; Gonçalves-Araujo et al., 2015; Mann et al., 2016).

The dependence of $a_\lambda$ on $\lambda$ is described using Equation (2):

$$a_\lambda = a_{\lambda 0} \, exp \, \{-S \, (\lambda - \lambda_0)\}, \tag{2}$$

where $a_{\lambda 0}$ is the absorption coefficient at reference wavelength $\lambda_0$, and $S$ is a spectral slope defining the spectral dependence of the absorption coefficient resulting from CDOM presence (Blough and Del Vecchio, 2002).

The spectral slope, $S$, indicates the rate at which the CDOM absorption coefficient decreases with wavelength increase (Carder et al., 1989). The value of $S$ varies with the source, aromatic content, and molecular weight of the CDOM (Blough and Del Vecchio, 2002; Helms et al., 2008; Granskog et al., 2012). In near-shore regions, which are under the influence of terrestrial sources with high concentrations of CDOM, $S$ values increase due to the conservative mixing of terrestrial CDOM (high $a_\lambda$, low $S$) with oceanic CDOM (low $a_\lambda$, high $S$) (Stedmon and Markager, 2003). Therefore, it is also widely accepted that the spectral slope $S$ can be used as a proxy for CDOM composition. However, its usefulness is limited by the fact that $S$ depends on the wavelength interval over which it is calculated (Carder et al., 1989; Stedmon et al., 2000). Following recommendations by Helms et al. (2008) a wavelength interval of 275-295 nm was chosen for detailed spectral analysis because it demonstrates the biggest variability of optical parameters when mixing waters with contrasting optical characteristics. The ratio of $S$ values from the shorter (275-295 nm) and the longer wavelength region (350–400 nm), termed the slope ratio, $S_R$, was calculated as described by Helms et al. (2008). $S_R$ values for terrestrial CDOM typically are <1 whereas oceanic CDOM and extensively photodegraded terrestrial CDOM are typically >1.5 (Stedmon and Nelson, 2015). Specific UV absorbance (*SUVA*), defined as the UV absorbance of a water sample at 254 nm normalized for DOC concentration, is used to estimate the degree of aromaticity in bulk CDOM (Weishaar et al., 2003):

$$C_{Ar} = 6.52*SUVA + 3.63, \tag{3}$$

where $C_{Ar}$ is the percentage of total carbon existing as aromatic carbon.

This equation is applicable for a wide range of aquatic environments (seas, bogs, lakes) since the authors used humic substances that have different chemical characteristics and demonstrated a strong correlation (r = 0.98) between the specific UV absorbance and aromatic carbon content (Weishaar et al., 2003).

Dissolved lignin is a well-established biomarker of terrigenous DOM in the ocean and has been successfully applied as a tracer of riverine inputs in the Arctic Ocean (Fichot et al., 2016). To assess its content in coastal waters two empirical models for the retrieval of the sum of nine lignin phenols (TDLP$_9$, nmol L$^{-1}$) from $a_\lambda$ were applied (Fichot et al., 2016). For $a_{250} < 4$ m$^{-1}$, a "low-CDOM" sub-model based on a simple linear regression was used,

$$\ln(TDLP_9) = 0.7672 \cdot a_{263} - 0.3987 \tag{4}$$

For $a_{250} \geq 4$ m$^{-1}$, a "high-CDOM" sub-model based on a multiple linear regression was used,

$$\ln(TDLP_9) = -2.282 \cdot \ln(a_{350} - 8.209 \cdot \ln(a_{275}) + 11.365 \cdot \ln(a_{295}) + 2.909 \tag{5}$$

CDOM fluorescence was measured with a WETStar DOM fluorometer (Table 1), which is suitable for in situ measurements without prior filtration of water (Belzile et al., 2006). The raw voltage from the fluorometer was converted to quinine sulfate units (QSU) (Belzile et al., 2006).

**3.4 Statistical treatment and graphical representation of the data**

Data were tested statistically using an empirical distribution function test in the Statistics 7.0 software package. Descriptive statistics were calculated for the 95 % confidence interval of the mean (P = 0.95, alpha = 0.05). Most of the plots and maps in this study were created with the Ocean Data View software (Schlitzer, 2011).

**4 Results and Discussion**

**4.1 Hydrometeorological situation and CDOM spatial variability**

The spatial distribution of sea-surface salinity obtained in September 2003, 2004, 2005, 2008, and 2011 is shown in Fig. 2a. During summertime, the coastal currents transport a large part of the Lena River water eastward into the East Siberian Sea. The overall hydrological conditions in the near shore zone were mainly determined by the interaction of river and marine water (Fig. 2a). The sea surface salinity showed a general eastward increasing trend. The salinity values varied between 3.1 (2004) and 32.34 (2011), with the lowest values associated with fresh water input from the Lena River and the higher values attributed to the presence of Pacific water. The maximum eastward spreading of river water was detected in September 2004 when the freshwater signal was found in the vicinity of Long Strait.

Comparative analysis of 2003, 2004, 2005, 2008, and 2011 data showed that, in general, CDOM concentration in ESAS surface waters varies depending upon distance from the river water source: the maximum CDOM was recorded near river mouths, and the minimum in regions remote from direct river discharge. However, the spatial distribution of waters with high CDOM values on the ESAS differed significantly in different years (Fig. 2b). The CDOM isoline equal to 15 QSU in Figure 2b can be used to compare the distribution of river water in different years. Location of this CDOM isoline correlates spatially with isohaline 24.5 which has been suggested as a marker for the boundary of surface shelf waters diluted by riverine runoff in the Siberian seas (Nikiforov and Shpaikher, 1980; Semiletov et al., 2000, 2005).

Comparison of the discharge of the main rivers flowing into the Laptev and East Siberian seas (the Lena and Kolyma rivers), which was based on Tiksi Hydromet data and the http://rims.unh.edu website, demonstrated that the maximum total discharge of the Lena and Kolyma rivers during the study period was recorded in 2008 (829 km$^3$) and the minimum in 2003 (543 km$^3$). However, the maximum distribution of ESAS waters with high CDOM concentrations was observed in 2004 and

the minimum was observed in 2011 (Fig. 2b). This indicates a complex and multifactorial correlation between river discharge and distribution of CDOM-enriched waters on the ESAS.

Following Proshutinsky et al. (2015), dominant atmospheric processes and differences in ice-coverage area mainly determine surface hydrology in the wind-driven ESAS. Therefore, we assume that inter-annual variability of the sea surface CDOM distribution on the ESAS is also determined by atmospheric processes.

National Centers for Environmental Prediction (NCEP) sea level pressure (SLP) data were employed to describe the atmospheric circulation over the Arctic Ocean (www.esrl.noaa.gov). The SLP fields, averaged over the summer season (July
– September) for each year, are shown in Figure 2c.

During the **2003** summer season cyclonic atmospheric circulation dominated over the central Arctic Ocean. SLP as low as 1005 mbar extended over the Laptev and the East Siberian seas (Fig. 2c). The development of a deep atmospheric depression caused onshore winds on the western periphery of the cyclone and transferred freshened shelf waters to the east (Pipko et al., 2008; Savel'eva et al., 2008). Waters with high CDOM concentrations spread to 165° E although river discharge was at a
minimum in 2003 (Fig. 2b). The September 2003 sea-ice extent was at a maximum from 2003 to 2011 in the Arctic Ocean ($6.1 \cdot 10^6$ km$^2$, http://nsidc.org/data/seaice_index/), but Dmitry Laptev Strait was clear of ice in early August; thus, intensive water exchange occurred between the Laptev and East Siberian seas under a prevailing west wind.

In **2004** the summer low pressure north of the East Siberian Sea was weaker while an anticyclone formed above the Canadian Arctic Archipelago (Fig. 2c). High river discharge, ice conditions, and offshore winds determined the maximum
distribution of river waters in the ice-free East Siberian Sea (Fig. 2b).

Predominance of cyclonic atmospheric circulation over the western Laptev Sea and anticyclonic circulation over the Beaufort Sea created significant SLP gradients and, thus, strong winds in **2005** (average wind speed of up to ~ 11.2 m sec$^{-1}$, Fig. 2c). Strong southeastern winds prior to the expedition hampered the spread of river waters north and east of the river delta (Fig. 2b), intensified the processes of thermoabrasion (Günther et al., 2013), and facilitated the release of eroded carbon
into the water. The volume of river discharge was the second largest in the period of research. This is the year when the maximum absolute CDOM value (over 100 QSU, compared to an average of 25 QSU during the study period) was recorded in Buor-Khaya Bay surface water.

In **2008**, the summer atmospheric pressure field was conditioned by the dominant anticyclone over the Beaufort Sea and a weak cyclone over Siberia, which caused southeastern winds over the ESAS (Fig. 2c). Thus, although the river discharge
was great (the Lena River maximum discharge occurred during the investigated period), freshened waters were located in the southeastern Laptev Sea and the western East Siberian Sea and did not penetrate into the eastern part of the sea. It should be noted that the ice extent was less in September 2008 compared with the previous study periods. The whole shelf was ice-free that year and the ice boundary moved north of 80° N in some areas (http://www.aari.nw.ru/).

An area of high SLP over the Arctic Basin and low SLP above the continent occurred in the summer season (July-August) of
**2011** (Fig. 2c). The averaged meridian wind speed was 1.5 m s$^{-1}$ and the averaged area wind speed was 3.5 m s$^{-1}$ (http://www.cdc.noaa.gov), which determined weak offshore east and southeast winds of 4 m s$^{-1}$. September sea-ice extent in the Arctic Ocean was smaller in 2011 than in 2008 (4.6 and 4.7 $*$ 10$^6$ km$^2$, respectively), but the ESAS ice edge was markedly closer to the median extent for 1981-2010 in 2011 (http://nsidc.org/data/seaice_index/). During the summer season, the East Siberian Sea ice edge did not reach 73° N and meltwater was transported from the East Siberian Sea to the
Laptev Sea under the influence of east and southeast winds. The fraction of meltwater depleted of CDOM reached 20 % (Anderson and Amon, 2015) in the surface waters of the Laptev Sea shelf in September 2011 (Pipko et al., 2015). The situation radically changed in September, when low SLP dominated over the Arctic Basin. The wind changed towards the north and northwest and became stronger. Low river discharge, weak winds in preceding summer months, and strong onshore winds during the study period provided for a weak spreading of river water and intensive mixing of seawater. High
meltwater content and a deeper mixed layer relative to September 2005 and 2008 (Pipko et al., 2016) also contributed to a

decrease of CDOM concentration. Eventually, the lowest absolute CDOM values and the minimum distribution area for CDOM-enriched waters were recorded in September 2011 (Fig. 2b).

Taken together, field data analysis showed that the prevailing type of atmospheric circulation and the position of action centers relative to each other were the dominant factors controlling the ESAS surface CDOM spatial distribution, while inter-annual variabilities of river discharge and ice extent were also significant. The greatest offshore propagation of waters with high CDOM content was recorded in 2004 when a cyclone was located above the central region of the Arctic Ocean basin and a high atmospheric pressure field formed over the Beaufort Sea. In September 2004 the East Siberian Sea surface salinity from Kolyma Bay (near $160^{\circ}$E) to Long Strait (roughly near $178^{\circ}$E) varied from 20 to 24, much lower than in other years (Semiletov et al., 2005), which is reflected in high CDOM values. Note that the prevailing type of atmospheric circulation and the position of atmospheric action centers are also dominant factors controlling variability in Siberian river discharge (Semiletov et al., 2000; Savelieva et al., 2000). Thus, this interplay between atmospheric forcing factors, riverine discharge, and redistribution of fresh water traced by CDOM values across the arctic seas should be studied further to achieve a better understanding of Arctic system complexity.

A significant negative correlation between CDOM and salinity in surface ESAS water (r = - 0.88, Fig. 3) in the salinity range from 3 to 24.5 indicates a terrigenous source for CDOM, and somewhat conservative mixing of CDOM between fresh water and seawater. Deviations of CDOM values from the trend line are determined by the presence of both meltwater depleted in CDOM (Anderson and Amon, 2015) and multiple riverine sources with varying DOM properties. At higher salinities (> 24.5) correlation in the surface layer between these parameters is practically absent (Fig. 3), and the distribution of CDOM throughout the water column is completely different in different ESAS areas (Fig. 4). Figure 4 illustrates two examples of representative salinity and CDOM profiles measured on the ESAS in September 2004 at station No.118 located near the Lena River mouth and at station No. 97 located in Long Strait, farther away from direct riverine influence. Note that a significant negative correlation between CDOM fluorescence and salinity was found for the whole water column data in the 2004 survey (r = - 0.84, N = 8920), indicating a predominantly riverine origin of CDOM. Inspection of these individual profiles shows a more complex situation (Fig. 4c). Each of the two profiles collected in the relatively shallow shelf water of the Laptev and East Siberian seas showed a significant correlation between CDOM and salinity, but these correlations had opposite signs. At the western station No.118 CDOM values decreased with higher salinity (r = - 0.99), while at the eastern station No.97 CDOM values were positively correlated with the salinity (r = 0.84). Thus, at higher salinity on the eastern ESAS, the presence of water masses with distinctive optical properties was responsible for the different behavior of CDOM.

## 4.2 Spatial variability of CDOM spectral characteristics

Spectral characteristics ($a_{350}$, $S_{275-295}$, $S_R$) that provided both quantitative and qualitative information about CDOM, were determined during field work in 2004, 2005, and 2011.

CDOM absorption spectra for September 2004 surface waters at two stations (stations 118 and 97) located in contrasting shelf zones (Fig. 1) are shown at Figure 5. Spectrum for station 60 located in the East Siberian Sea (in a moderate zone of mixing river and ocean waters) is also presented. CDOM absorption spectra measured for different waters have substantially different levels of absorption. Despite that CDOM dominates the absorption spectra in the blue and UV wavebands, the absorption coefficient $a_{350}$ for these stations demonstrated considerable spatial variability, from 0.7 $m^{-1}$ (station 97) to 11.1 $m^{-1}$ (station 118). Values obtained in the area of maximal Lena River influence coincided well with data given in Walker et al. (2013) for a mid-flow regime of the Lena River (about 13.1 $m^{-1}$). It was shown that during all flow regimes $a_{350}$ was highest in the Lena River relative to other large Arctic rivers (Walker et al., 2013).

The relationship between river water $a_{350}$ values and river discharge volume (Walker et al., 2013) also could be traced in the ESAS surface waters. According to our data, the $a_{350}$ values also increased in the shelf waters with an increase in riverine runoff (Figs. 2b, 6).

The multi-year surface water $a_{350}$ distribution showed the same feature for each year: the highest $a_{350}$ values were observed in the Laptev Sea waters under direct Lena River influence, with a decrease in $a_{350}$ toward the saltier waters of the Chukchi Sea (Fig. 6).

The $S_{275-295}$ values and spectral slope ratio $S_R$ correlated well with both DOM molecular weight and source (Helms et al., 2008). The lowest $S_{275-295} = 14.7$ μm$^{-1}$ in September 2004 was found near the river delta in the Laptev Sea (Fig. 5). This value is in the same range as previously reported values characterizing the surface waters of the eastern Arctic seas, which are dominated by allochthonous DOM during the late summer season (Stedmon et al., 2011; Walker et al., 2013; Gonçalves-Araujo et al., 2015). This value is also comparable to values found in other rivers studied in the Pacific sector of the Arctic. For example, $S_{275-295}$ reaches 12 μm$^{-1}$ in the Yukon River during the peak flood periods when the concentrations of DOM and its colored fraction are the greatest (Spencer et al., 2009). $S_{275-295}$ increases to 18.9 μm$^{-1}$ in the western East Siberian Sea where river influence decreases. $S_{275-295}$ further increases to 21 μm$^{-1}$ (Fig. 5) in Long Strait where primary production is high (Walsh et al., 1989) and the main portion of DOM is produced in situ. Overall, this value was significantly lower than values observed for open ocean waters, which are typically 25-30 μm$^{-1}$ (Blough and Del Vecchio, 2002). The values characteristic for open seawater (28-29 μm$^{-1}$) were found in the east of the ESS in September 2011, when a weak spreading of the riverine waters was detected.

$S_R$ was calculated for the shallow ESAS characterized by different intensities of terrigenous influence. The lowest values were obtained in the southeastern Laptev Sea (0.78) and the maximum $S_R$ (4.0) was found in the area influenced by the highly productive Pacific-origin waters in Long Strait where the optical properties are determined by phytoplankton-derived autochthonous CDOM with low molecular weight and lower amounts of aromatic functional groups (Fig. 6). High values of $S_R$ in the surface waters of the eastern East Siberian Sea confirm conclusions about weak spreading of riverine waters on the ESAS in 2011 and point to effective transport of Pacific-origin water on the ESAS under the anticyclonic type of atmospheric circulation (Fig. 2).

Figure 6a demonstrates the spatial distribution of aromatic carbon ($C_{Ar}$) in ESAS surface water in September 2004. The corresponding calculations show that the content of aromatic carbon within DOM remains almost constant in the region of substantial river impact to ~ 165° E (Fig. 6a). In the eastern part, a gradual decrease in aromaticity percentage was observed indicating the contribution of Pacific-origin waters, where allochthonous DOM with predominantly aliphatic character and much lesser absorption capabilities predominates.

## 4.3 Optical characteristics of the ESAS biogeochemical provinces

As was shown in previous studies, the concentrations of CDOM vary significantly within the ESAS (Belzile et al., 2006; Gonçalves-Araujo et al., 2015; Stedmon et al., 2011; Walker et al., 2013), mainly because of variations in the sources of DOM. The current multi-year investigations of the ESAS extend previous studies showing not only the inhomogeneity of spatial DOM variability, but also temporal DOM variability and its spectral characteristics distribution in the shallow shelf waters.

We found a stable tendency to a reduction in the concentrations of CDOM and aromaticity, and an increase of $S_R$ and $S_{275-295}$ values in the eastern (and northern) direction from the delta of the Lena River, which is the main supplier of dissolved OM to the eastern Arctic shelf. Our comprehensive data set made it possible to distinguish two ESAS provinces with contrasting biogeochemical regimes based on their spectral characteristics (Table 2). We used an approach described by Semiletov et al. (2005) and Pipko et al. (2005), which shows that, based on the hydrological and hydrochemical data distribution, two biogeochemical provinces (areas) could be identified in the shallow ESAS; a western area (heterotrophic biogeochemical province) that is influenced strongly by freshwater flux and transport of coastally-eroded particulate material enriched by OC, and an eastern area (autotrophic biogeochemical province) which is under the influence of highly productive Pacific-derived waters. A salinity of 24.5 was the conventional boundary of the western area. The spectral characteristics of CDOM

were significantly correlated with salinity in this part of the ESAS; correlation coefficient between $a_{350}$ and salinity for three years combined was -0.81. The relationship indicates that the conservative mixing of riverine and ocean waters plays a dominant role in regulating the variability of their values in this area. A similar pattern was established for the distribution of CDOM fluorescence values (Fig. 3). There was no relationship between optical parameters and salinity for the eastern part of the ESAS (Fig. 3); correlation between spectral indices and salinity was also absent in the eastern part with correlation coefficient between $a_{350}$ and salinity at -0.16 and between $S_R$ and salinity at 0.02. From year to year, depending on the dominant wind regime, the longitudinal boundary between western and eastern areas may shift by 10 degrees longitude and more.

Obtained data allowed us to calculate the lignin content in surface waters of the inner and middle ESAS based on CDOM spectral characteristics (Fig. 7) and to estimate the average values in each of the provinces. The most significant difference in the average values which characterize the western and eastern shelf regimes were found for the dissolved lignin concentration (Table 2). Lignin is a well-established biomarker of terrigenous OM in the ocean (Hedges et al., 1997; Fichot and Benner, 2014; Fichot et al., 2016). It is exclusively produced on land by vascular plants. As a result, lignin extracted from seawater and marine sediments has long been used to derive qualitative and quantitative information about the origins, transformations, and fates of terrigenous OM in the ocean (Amon et al., 2012; Fichot and Benner, 2014; Tesi et al., 2014; Fichot et al., 2016). The three large Siberian rivers, Lena, Yenisei, and Ob, which also have the highest proportion of forests within their watersheds among the six largest Arctic rivers, contribute about 90% of the total lignin discharge to the Arctic Ocean; the Lena River alone contributes about 48 % of the total annual lignin discharge into the Arctic Ocean (Amon et al., 2012). The calculated dissolved lignin concentration on the west ESAS in the late summer – fall season ranged from 11.9 – 263.2 nmol L$^{-1}$ with a mean of 89.5 nmol L$^{-1}$, indicating a strong influence of continental runoff on the shelf waters. The obtained values were much higher than lignin concentrations in the surface waters of the outer shelf and the continental slope of the Laptev and East Siberian seas (Fichot et al., 2016; Kaiser et al., 2017), where the influence of river flow is much weaker. The maximum concentrations of dissolved lignin were found in the shelf waters near the mouth of the Lena, Indigirka, and Kolyma rivers (263, 101, and 96 nmol L$^{-1}$, respectively) (Fig. 7); the observed values were significantly lower than the concentrations measured in the main streams of these rivers in the late summer-fall season (Mann et al., 2016). According to the obtained data, the concentrations of dissolved lignin decreased from the mouth of the Lena River to the mouths of the Indigirka and Kolyma rivers; thus, we can trace the signature of each river on the shelf from the measured spectral characteristics, as well as from the parameters calculated on the basis of those characteristics. Thus, at salinities equal to ~ 10, in the shelf waters near the river deltas, concentrations of dissolved lignin decrease in the eastern direction, from 180 nmol L$^{-1}$ near the Lena Delta to 101 and 85 nmol L$^{-1}$ near the mouths of the Indigirka and Kolyma, respectively. This tendency agrees well with the distribution of lignin phenol concentrations in Siberian river waters (Mann et al., 2016).

In eastern ESAS surface waters (Pacific-influenced province) the average dissolved lignin concentration decreased by more than an order of magnitude compared to the western part (from 89.5 to 5.1 nmol L$^{-1}$, Table 2). This indicates a minor effect of the terrigenous OM source on the biogeochemical regime of eastern ESAS waters. Together with the distribution of other optical parameters, the dynamics of lignin content indicate the "marine" DOM origin in the eastern ESAS and a transition from a heterotrophic to an autotrophic water regime over this part of the shelf. Note that the obtained values were slightly lower than the concentrations of dissolved lignin calculated for the surface water of the deep parts of the Laptev and East Siberian seas, where river water penetration is only possible under certain atmospheric circulation conditions (Bauch et al., 2011; Pipko et al., 2015, 2017; Kaiser et al., 2017).

The presented values of CDOM spectral parameters are comparable to findings from other studies previously conducted in this region (Stedmon et al., 2011; Walker et al., 2013; Gonçalves-Araujo et al., 2015; Mann et al., 2012, 2016). However, our data allow us to trace the dynamics of CDOM spectral characteristics in a highly dynamic shallow region of the Arctic

shelf, between heterotrophic riverine waters (Stedmon et al., 2011; Walker et al., 2013; Gonçalves-Araujo et al., 2015; Mann et al., 2012, 2016) and autochthonous seawater (Fichot et al., 2016; Kaiser et al., 2017).

## 4.4 Rapid assessment of dissolved organic carbon based on optical characteristics of dissolved organic matter

Conventional methods for the analysis of DOC are restricted to measurements of discrete samples and are limited to providing synoptic coverage on relatively small spatial scales. Estimating DOC concentrations via measuring optical DOM properties (absorption and fluorescence) therefore represents a compelling alternative (Fichot and Benner, 2011).

Methods to predict DOC concentrations from absorbance characteristics have been attempted since the early 1970s (Banoub, 1973; Lewis and Tyburczy, 1974). However, a prerequisite for successfully predicting DOC concentration is that the non-

absorbing DOC is at a constant or low level (Ferrari and Dowell, 1998). The processes responsible for DOC and CDOM distribution in the open ocean are typically independent, and the two pools usually demonstrate a negative rather than a positive correlation (Coble, 2007). The situation is different in shelf areas where terrestrial discharge is strong, and distribution of river water and seawater controls the distribution of both DOC and CDOM. A novel method to accurately retrieve DOC concentrations from CDOM absorption coefficients for the salinity range of 0-37, typically encountered in

river-dominated ocean margins, was successfully developed by Fichot and Benner (2011).

Here we present a method to estimate DOC concentration from in situ CDOM fluorescence using data collected in September 2004 and 2008 in surface waters of the shallow ESAS, where the Lena River is the main riverine source.

DOC and CDOM concentrations were measured simultaneously in the surface shelf waters of the Laptev and East Siberian seas during the 2004 and 2008 surveys to determine relationships between the two parameters.

In 2004, along with the study of filtered seawater sample absorption coefficients, CDOM was measured in situ using a WETStar fluorometer. Strong positive correlation (r = 0.98) was observed between $a_{350}$ measured in the filtered samples and CDOM of the unfiltered samples measured simultaneously in the ESAS surface water at 46 stations (Fig. 8).

This confirms previous results based on data obtained exclusively in the East Siberian Sea (Belzile et al., 2006) showing that the presence of suspended matter in water samples has little impact on CDOM values even in regions of intensive

terrigenous discharge. We have now compared the relationship between CDOM and $a_{370}$, which quantifies CDOM, using the East Siberian Sea data reported by Belzile et al. (2006) and our data obtained in the Laptev and East Siberian seas. A good agreement was found between Belzile's and our data (Fig. 8b). The correlation coefficient between CDOM and $a_{370}$ have been found to be the same in the ESAS (r = 0.97, N = 92) and in the Belzile et al. samples from Boothbay Harbor, West Harbor Pond, the Beaufort Sea, and the East Siberian Sea (r = 0.98, N = 74).

The relationship between DOC and CDOM showed a strong positive correlation for the 2004 and 2008 cruises (r = 0.99 for each survey, Fig. 9). We combined the two data sets to obtain a general equation connecting these parameters and to explore our capability to predict DOC concentration from the linear relationship between DOC and CDOM in the studied season.

An important implication of the obtained total relationship between DOC and CDOM (Fig. 9d) is the capability to constrain the DOC concentration for the ESAS surface waters in the late summer season using information about in situ CDOM

fluorescence. It allows DOC values to be calculated with a high spatial resolution based on direct fluorescence measurements, avoiding various artifacts induced by filtering and storing samples and, thus, adding to the limited available data on DOC distribution over the ESAS.

Overall, this simple empirical model provides a practical means to derive reasonably accurate estimates of DOC concentrations from CDOM in coastal waters of the Siberian Arctic shelf (Fig. 9, 10). The model adequately reproduces the

most important features of the measured DOC distribution in the shelf waters. A comparison of the measured and calculated DOC indicates that the model restored DOC values within ±10% of the measured values (Fig. 9). The % error was significantly higher than 10% only for two points in Long Strait, where influence of the Pacific-origin water was significant

(Fig. 9d). The model might also slightly overestimate some low DOC concentrations found in water with increased salinity (in the vicinity of Long Strait).

Therefore, this rapid assessment of DOC concentrations on the ESAS using a WETStar fluorometer is an effective tool for obtaining information on DOC distribution in summer seasons. This approach is reliable over the salinity range of 3 to 24.5; the lower limit is defined as the lowest conventional salinity level at which all the major marginal filters have already been passed (Lisitsyn, 1994) and the upper limit is a border of riverine water distribution (Nikiforov and Shpaikher, 1980).

## 5 Conclusions

A multi-year study of DOM optical parameters and the spectral characteristics of the DOM chromophoric fraction was carried out repeatedly probing the summer-fall season on the broadest and shallowest shelf in the World Ocean, the vast ESAS, from the Lena River delta in the Laptev Sea to Long Strait in the East Siberian Sea. For the first time, CDOM/DOC interannual variability in connection with atmospheric pressure fields and wind-driven water circulation was considered. The atmospheric circulation regime is the dominant factor controlling hydrography and spatial expansion of the area of

freshwater influence which determines CDOM/DOM spatial distribution on the ESAS.

The dynamics of DOM optical properties provided a new insight into biogeochemical processes in the ESAS. The spectral characteristics of CDOM were applied to identify two clearly distinct biogeochemical provinces in the surveyed area. The analysis of CDOM spectral characteristics has clearly shown that the major part of the Laptev and East Siberian Sea shelf is influenced by terrigenous DOM with high aromaticity and high lignin content, transported by riverine discharge.

The content of aromatic carbon within DOC remains almost constant in the western ESAS, which is the region of substantial river impact, while in the eastern ESAS, a gradual decrease in aromaticity percentage was observed indicating an increasing contribution of Pacific-origin waters, where allochthonous DOM with predominantly aliphatic character and much lesser absorption capabilities prevails.

We found a stable tendency of CDOM and dissolved lignin concentrations to be reduced and $S_R$ and $S_{275-295}$ values to
increase in the eastern direction from the delta of the Lena River, which is the main supplier of dissolved OM to the Eastern Arctic shelf.

The strong correlation between DOC and CDOM concentrations in surface shelf waters that are influenced by terrigenous discharge makes it possible to calculate DOC content from CDOM values assessed in situ using the WETStar fluorometer. Moreover, the reliable estimation of optical characteristics of coastal waters is crucial for validating and calibrating remote
sensing data processing results. Employing optical techniques can increase the temporal and spatial coverage of DOM measurements across the ESAS and help to more accurately estimate the amount of terrigenous DOM; this estimation is necessary for understanding how carbon budgets and fluxes will be altered under future climate change scenarios.

*Data availability.* DOM and all data will be made publicly available at the open-access Stockholm University Bolin Centre Database (http://bolin.su.se/data/), once the manuscript is published.

**Acknowledgments**

This work was supported by the Russian Government (grant 14.Z50.31.0012), the Far Eastern Branch of the Russian Academy of Sciences (FEBRAS); the International Arctic Research Center (IARC) of the University of Alaska Fairbanks through NOAA Cooperative Agreement NA17RJ1224; the U.S. National Science Foundation (Nos. OPP-0327664, OPP-0230455, ARC-1023281, ARC-0909546); and the NOAA OAR Climate Program Office (NA08OAR4600758). S.P. and I.P.
thank the Russian Foundation for Basic Research (RFBR, No. 14-05-00433a); E.S and I.V.P acknowledge RFBR grants No. 15-05-09284 and 16-04-01753. N.S. and A.R. acknowledge the Russian Science Foundation (grant No. 15-17-20032). O.G. acknowledges support from the Swedish Research Council, the Knut and Alice Wallenberg Foundation, and an ERC

Advanced Grant (ERC-AdG CC-TOP project #695331). I.S. and N.S. acknowledge support from the ICE-ARC EU FP7 project. We thank Ronald Benner for DOC measurements in water samples taken in the ESAS onboard HV *Ivan Kireev* in 2004 and two anonymous reviewers for their constructive comments. We thank Candace O'Connor for English editing.

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

**Table 1.** Dates, number of stations, samples collected, and parameters measured in the surface layer during cruises.

| Dates of cruises | Region | Number of stations | Parameters measured |
|---|---|---|---|
| HV *Ivan Kireev*, 7 - 21 Sept. 2003 | East Siberian Sea | 41 | CTD with CDOM ($n = 41$) |
| HV *Ivan Kireev*, 31 Aug. - 15 Sept. 2004 | Southeastern Laptev Sea, East Siberian Sea | 92 | CTD with CDOM ($n = 92$), $a_\lambda$ ($n = 46$), DOC ($n = 20$) |
| MV *Auga* 14 – 25 Sept. 2005 | Southeastern Laptev Sea, southwestern East Siberian Sea | 64 | CTD with CDOM ($n = 64$), $a_\lambda$ ($n = 32$) |
| HV *Yakob Smirnitskiy*, 8 Aug. – 18 Sept. 2008 | Eastern Laptev Sea, East Siberian Sea | 1766 (SWI) | CTD with CDOM ($n = 1766$), DOC ($n = 12$) |
| RV *Academician M.A. Lavrentyev*, 15 Sept. – 4 Oct. 2011 | Eastern Laptev Sea, East Siberian Sea | 89 | CTD with CDOM ($n = 89$), $a_\lambda$ ($n = 21$) |

$n$ is the number of measurements

**Table 2.** DOM optical characteristics, dissolved lignin, and salinity in two biogeochemical provinces of the inner and middle ESAS surface water (obtained during the 2004, 2005, and 2011 surveys).

| Parameter | Absorption coefficient ($a_{350}$), m$^{-1}$ | | | | | $S_{275\text{-}295}$, µm$^{-1}$ | | | | | $S_R$ | | | | | CDOM, QSU | | | | | Dissolved lignin ($TDLP_9$), nmol L$^{-1}$ | | | | | Salinity | | | | |
|---|---|---|---|---|---|---|---|---|---|---|---|---|---|---|---|---|---|---|---|---|---|---|---|---|---|---|---|---|---|---|
| Region | AVG | Md | STD | Min | Max | AVG | Md | STD | Min | Max | AVG | Md | STD | Min | Max | AVG | Md | STD | Min | Max | AVG | Md | STD | Min | Max | AVG | Md | STD | Min | Max |
| Western province $n=90$ | **5.2** | 4.25 | 3.05 | 0.5 | 17.3 | **17.6** | 17.5 | 1.7 | 10.4 | 21.1 | **1.1** | 1.0 | 0.2 | 0.8 | 1.8 | **43.4** | 37.4 | 28.5 | 3.4 | 135.9 | **89.5** | 73.4 | 53.0 | 11.9 | 263.2 | **14.4** | 15.7 | 5.3 | 0.1 | 24 |
| Eastern province $n=9$ | **0.6** | 0.6 | 0.4 | 0.1 | 1.4 | **22.6** | 23.3 | 4.7 | 13.1 | 29.0 | **2.9** | 3.2 | 1.5 | 0.8 | 4.8 | **1.6** | 1.1 | 1.4 | 0.4 | 5.1 | **5.1** | 4.6 | 3.0 | 1.6 | 10.8 | **28.0** | 28.1 | 2.7 | 24.7 | 32.1 |

AVG – average value, Md – median value, STD – standard deviation, $n$ is the number of measurements.

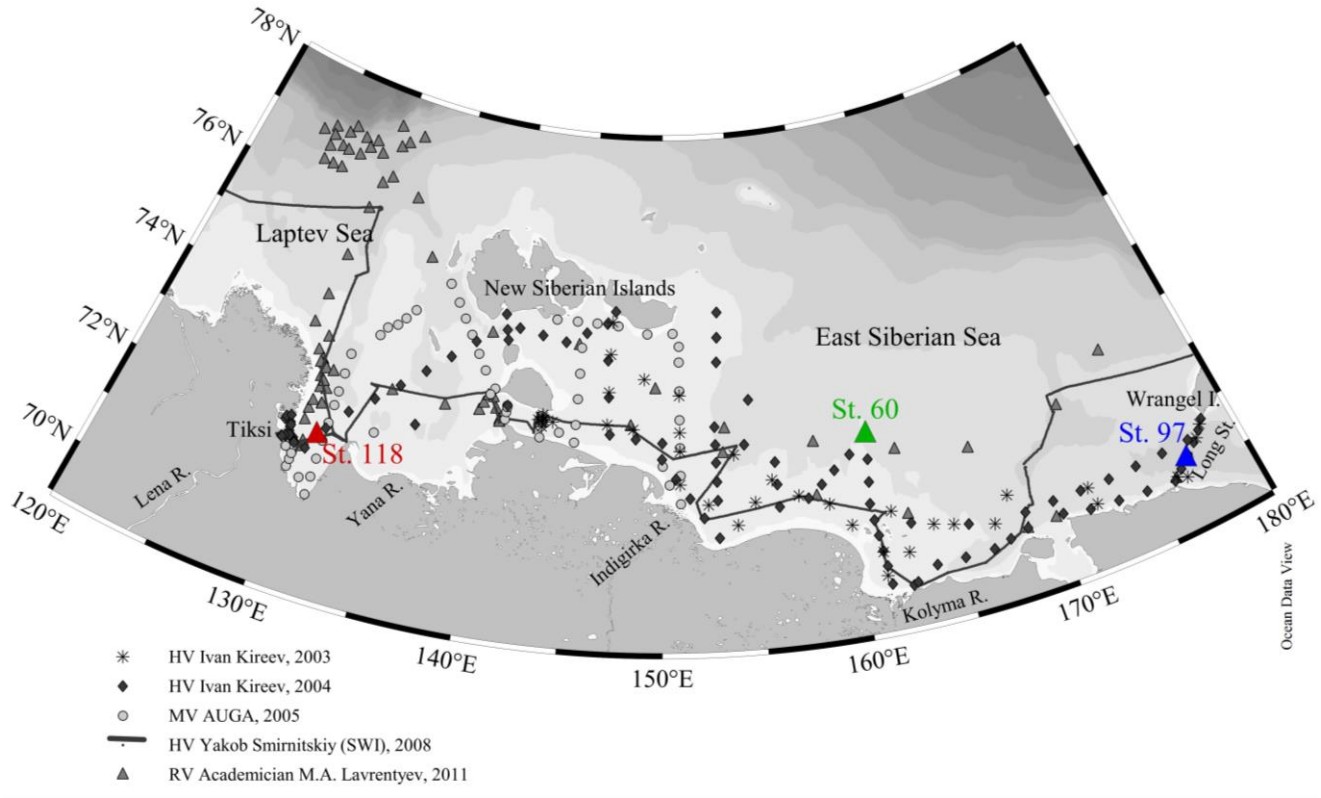

**Figure 1.** Study area: Location of oceanographic stations accomplished in August-October 2003, 2004, 2005, 2008, and 2011. Location of selected stations on the ESAS in September 2004 is shown in color.

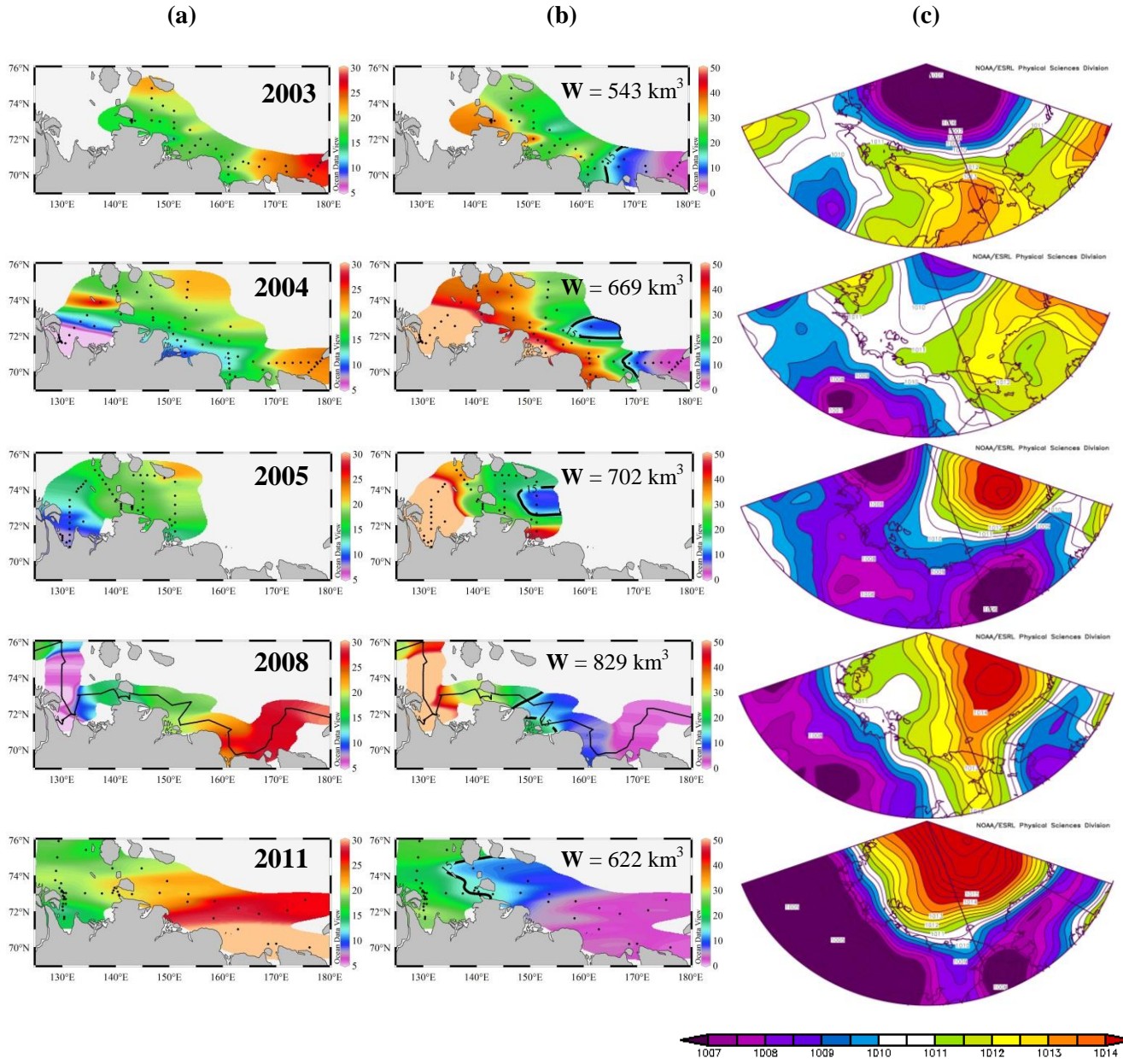

**Figure 2.** Distributions of surface salinity (**a**) and CDOM (**b**) in September 2003, 2004, 2005, 2008, and 2011, and (**c**) sea level pressure fields (mbar), averaged over the summer season (July–September) from NCEP data. **W** - total annual river discharge.

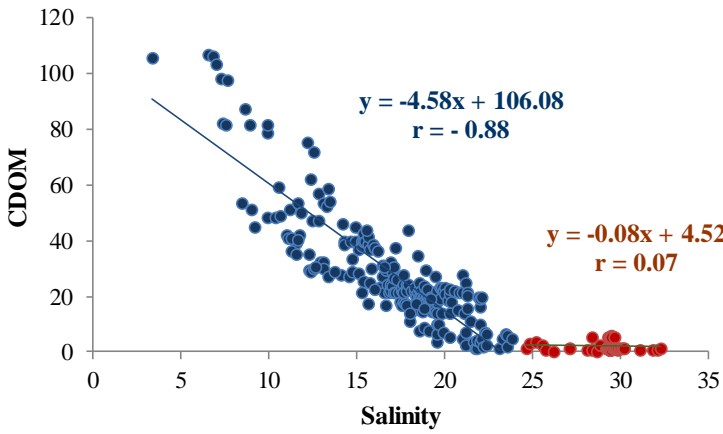

**Figure 3.** The relationship between the salinity and CDOM (QSU) in the ESAS surface water in August-September 2003, 2004, 2005, 2008, and 2011 (blue circles – salinity < 24.5, red - > 24.5).

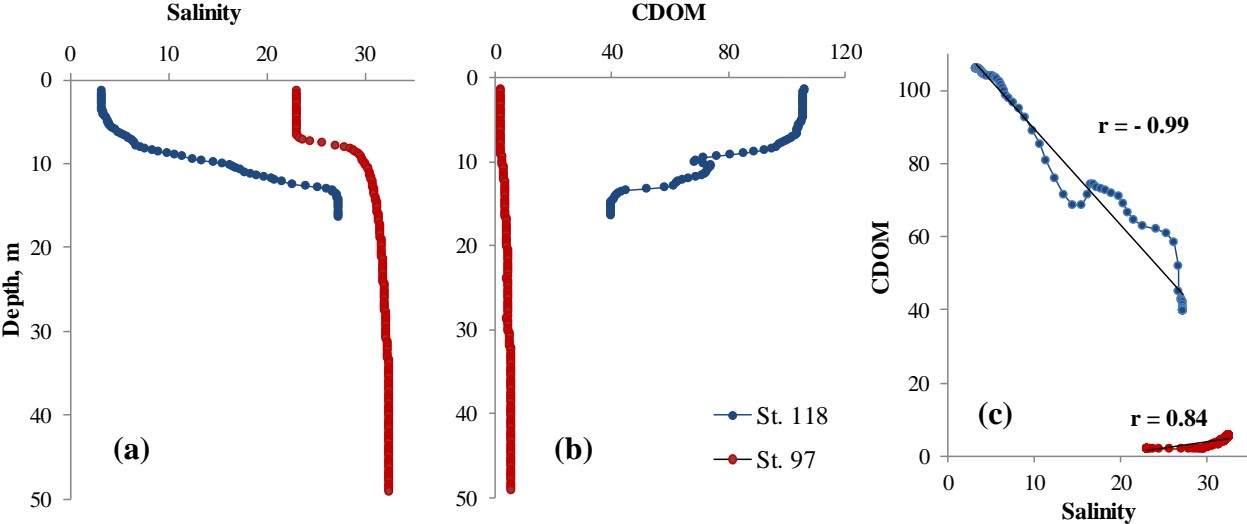

**Figure 4.** Depth profiles of salinity (**a**) and CDOM (QSU) (**b**) measured at two stations on the ESAS, and (**c**) relationship between CDOM and salinity, September 2004; see Fig. 1 for location of stations.

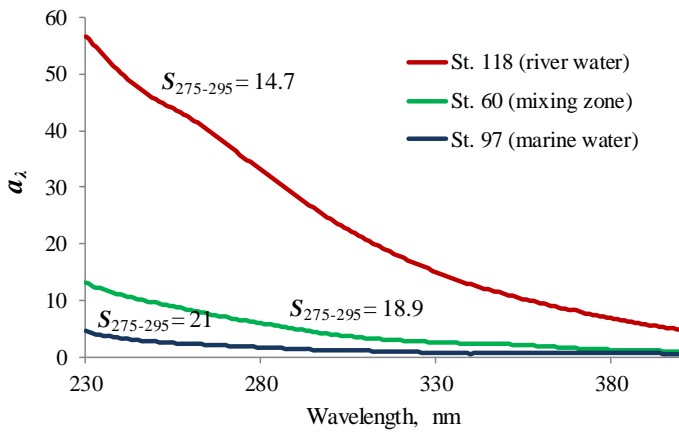

**Figure 5.** Surface CDOM absorption coefficient spectra $a_\lambda$ (m$^{-1}$) for station (St.) 118, 60, and 97, September 2004; see Fig. 1 for location of stations.

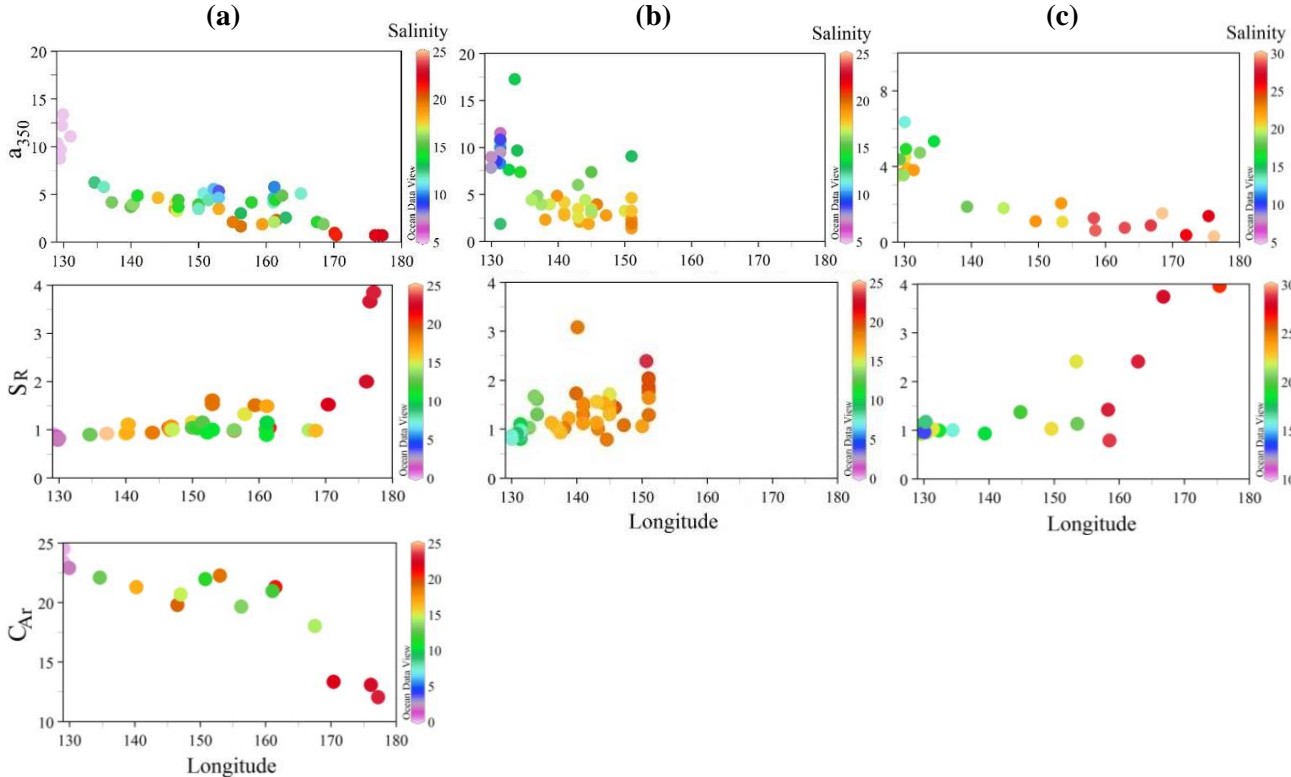

**Figure 6.** The relationship between CDOM absorption coefficient at the 350 nm wavelength ($a_{350}$, m$^{-1}$), $S_R$ and the aromatic carbon content ($C_{Ar}$, %) and salinity in the ESAS surface waters, September 2004 (**a**), 2005 (**b**), and 2011 (**c**).

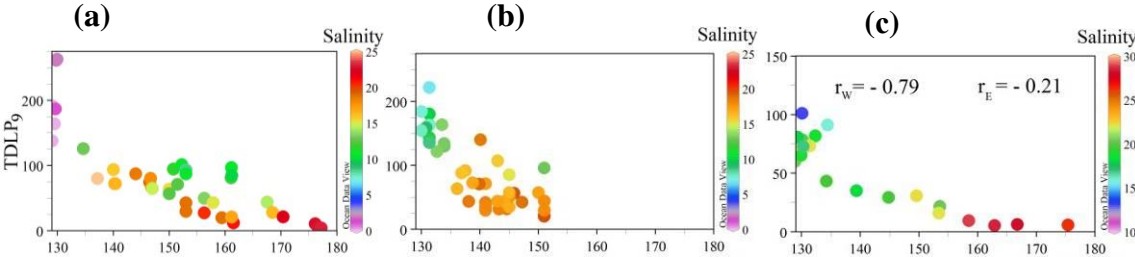

**Figure 7.** The relationship between the dissolved lignin concentrations (TDLP$_9$, nmol L$^{-1}$) and salinity in the ESAS surface waters, September 2004 (**a**), 2005 (**b**), and 2011 (**c**), where r$_W$, r$_E$ represent correlation coefficients between TDLP$_9$ and salinity for western and eastern parts of the ESAS, respectively, for these three years combined.

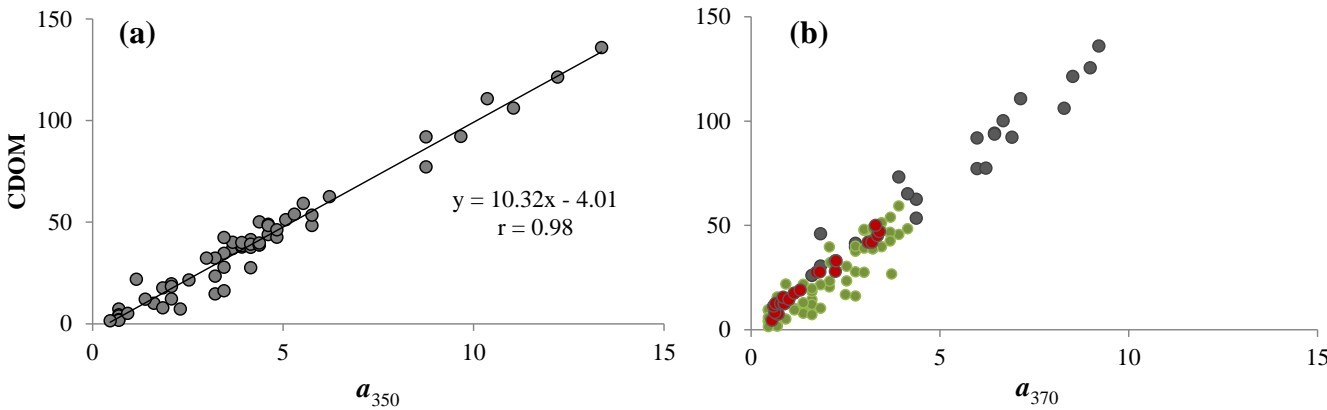

**Figure 8.** Correlation between CDOM (QSU) and absorption coefficient at 350 nm ($a_{350}$, m$^{-1}$) in ESAS surface water (**a**) and relationship between CDOM (QSU) and absorption coefficient at 370 nm ($a_{370}$, m$^{-1}$) in water column (**b**), September 2004. Grey – our data in the Laptev Sea (N = 23), green – our data in the East Siberian Sea (N = 69), and red – data from Belzile et al., 2006 (N = 23).

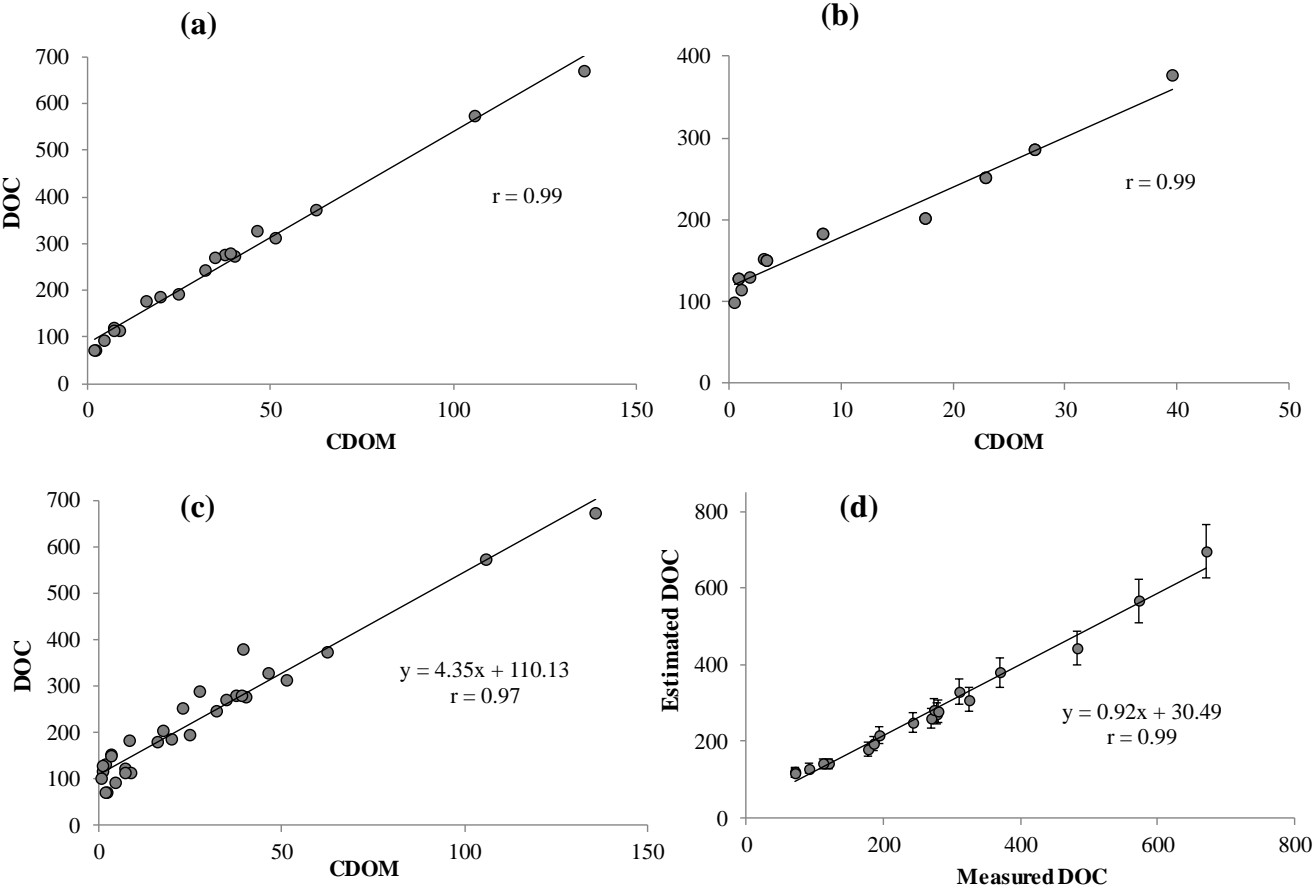

**Figure 9.** DOC concentration (µM) versus CDOM (QSU) in the ESAS surface water, September 2004 (**a**), 2008 (**b**), and combined for two years (**c**); (**d**) estimated DOC concentration against measured DOC concentration with 10% error bars.

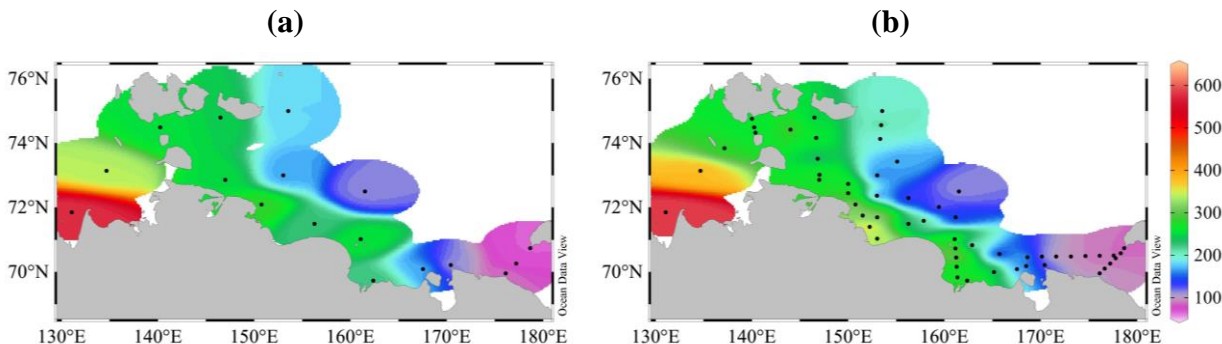

**Figure 10.** DOC distribution (µM) in the ESAS surface waters in September 2004: (**a**) measured DOC concentrations, (**b**) concentrations of DOC estimated from CDOM.