# Peer review of "Dissolved organic matter and its optical characteristics in the Laptev and East Siberian seas: Spatial distribution and inter-annual variability (2003-2011)"

_Ocean Science, 2017_

## Referee Comment (RC1) · Anonymous Referee #1 · 12 May 2017

**General Comments:**

In the manuscript entitled "DOM and its optical characteristics in the Laptev and East Siberian seas: Spatial distribution and inter-annual variability (2003-2011)" the authors describe the dominant factors which control the distribution of dissolved organic matter (DOM) in the Siberian shelf seas. On the basis of a data set of several years the authors also try to explain the reason for the observed year-to-year variability of the DOM distribution. A further focus of the work is on the estimation of the utility of in situ fluorescence measurement.

Basically, the article describes phenomena that have been already investigated and published by other authors, i.e.: - The atmospheric forcing of the Lena freshwater plume - and thus the associated high content of terrestrial DOM (for example: Dmitrenko et al., 2005, Wind-driven surface hydrography of the eastern Siberian shelf, doi:10.1029/2005GL023022).- The geochemical behavior of DOM on the Laptev and East Siberian Seas (for example: Alling et al., 2010, cited in the manuscript) - The usefulness of in situ fluorescence measurements for the investigation of DOM in the East Siberian and Laptev Seas (Belzile et al., 2004; cited in the manuscript). Actually, some of the co-authors of this manuscript were also co-authors of the study published by Belzile. Because both studies are based on samples from the same region and the same year, the question is whether it is the same set of samples? What is the main difference to the work of Belzile et al.?

In general, it is hard for me to recognize on which data set the analysis is based on. According to table 1, CDOM (for absorbance measurement) was sampled on 245 stations, but in Table 2 (statistics) the number of samples (N) is 90. Also Figure 2a and 5 give no explanation because the gridded maps do not show the sampling locations for CDOM. Figure 9 (DOC) only shows  $\sim$ 19 sampling locations. This makes it difficult to follow the study's line of argumentation - at least for me. In my opinion, the presentation of the data has to be revised.

The authors distinguish a western biogeochemical province from an eastern province and state that the CDOM concentration is high near the Lena Delta and low in the eastern East Siberian Sea, a region that is less influenced by riverine freshwater. They also found a strong negative correlation between the CDOM absorption and the salinity and concluded that the CDOM is mainly of terrestrial origin. For the Arctic Ocean, this is a well-documented hypothesis (see the work of R. Amon, C. Stedmon; M. Granskog and many others). Unfortunately, no figures were presented to show the correlation between salinity and CDOM. In my opinion, we are not dealing here with different biogeochemical regimes but with differences in the hydrography and spatial expansion of Interactive comment

the region of freshwater influence (ROFI) driven by atmospheric forcing. The conclusion that "the atmospheric circulation regime is the dominant factor controlling CDOM spatial distribution" is therefore somewhat misleading. Are the geochemical processes fundamentally different within and outside of the ROFI? This point needs to be discussed in much greater detail

Specific and technical comments:

- Are the DOM data accessible? - Please indicate the sampling locations in the figures. - I guess you used the DIVA tool (ODV) for the spatial gridding. I believe the interpretation of data can be misleading if a spatial interpolation that is based on 17 sampling locations covers an area of approx. 500.000 square kilometers (Figure 9.). The DIVA interpolation gives DOC and CDOM concentrations ~400 km north of the last sampling location. I would like to suggest to redraw all figures and to show the original data (e.g. as colored dots) instead of the interpolated data. - Samples were taken from a "seawater intake system" that pumped the water into a 300 l barrel. Have you measured the salinity directly in the sample, or did you use the salinity data from the Seabird CTD? The inlet of the ship was at 4m water depth. If you have taken the salinity data from the CTD, which depth have you chosen? - Please do not use acronyms in the heading of the manuscript - Page 3 line 147: I guess 0.7  $\mu$ m is the correct nominal pore size of the filters. - Is figure 3 really necessary? What does it tell about the CDOM distribution? - Figure 4: Wouldn't it be better to show the relation between aromaticity and salinity instead of longitude? - English is not my first language, but I believe the text needs a linguistic revision.

OSD

---

## Referee Comment (RC2) · Anonymous Referee #2 · 21 May 2017

Review of Svetlana et al DOM and its optical characteristics in the Laptev and East Siberian seas: Spatial distribution and inter-annual variability (2003-2011).

The manuscript has the aims of reporting on the inter-annual variability in CDOM and DOC in the Laptev and East Siberian sea. It reads very much like a cruise report and would benefit from a more comprehensive data analysis and discussion of the results obtained. The referencing of previous literature is suboptimal and at times inappropriate. In my opinion there is a missed opportunity for a solid analysis on the linkage between CDOM absorption, fluorescence and DOC across several years. The

authors are in possession of a unique dataset which is only lightly touched on. Why not take more inspiration from the Belzile paper cited and include a comparison of your data with theirs from the East Siberian sea? Does the same FDOM to CDOM relationship exist? The section on the inter-annual variability is difficult for the reader to follow as is. It would likely be easier if figure 5 and 6 were combined so that the sea level pressure maps could be compared with the CDOM and salinity distribution maps. Alternatively, the authors could just compare the maps of both salinity and SLP, then in a separate figure reveal how robust the salinity CDOM relationship was. In this form the manuscript is not suitable for publication and I recommend re-submission after revising the data analysis. Whilst doing this you should consider splitting the results and discussion sections to allow for a better separation between your results and reflections on how your findings link to other studies.

Other points to address:

Line 8. "amount" rather than "volume".

Line 21. Replace "were" with "was"

Try to avoid use of "e.g." in referencing and citing very many studies. Find the most relevant and limit it to 3-4.

Line 46. Replace "gives input" with "supplies".

Line 50-51. I suggest you specify this more. Many rivers and streams have high or higher DOC but few large rivers have concentrations this high at their mouth.

Line 54. Replace "lead" with "is leading to"

Line 56-68. This section should be rephrased and better references found. If you do not want to have too many references I recommend you pick either the original papers or first to demonstrate this in the Arctic. Currently there is a bizarre selection of studies cited and not all directly relevant.

Line 78-79. Several of these references are not even Arctic. Line 80. Were there not any additional scientific aims or hypothesises? Possibly developed during the data analysis for this study? Try to mention them here. As stated now the aim reads very much as a data report.

Delete line 83-86. This has been established in the Introduction.

Line 95. Check your phrasing of " would be oxidised".

Line 133-134. Delete this. It is a standard fluorometer which is readily available. No need for this. Also the description of the interior optics can be removed. Not really necessary and appears to be copy pasted word for word from Belize et al 2006 paper, which is a little alarming.

Line 146. What ranges? I do not understand.

Lin 150. It is not valid to apply the fit across this range. The spectrum does not behave exponentially and in many samples there will be a shoulder at 280. Additionally the absorption below 240 will be mainly due to other constituents.

Line 156. Sr is not explained, and the whole this part if poorly written.

Line 157. SUVA is not that recent and include a citation of the original paper for this (Weishaar ). Line 158. I do not agree with this sentence. Starting "The last parameter..."

Line 161. I do not agree with this extrapolation. The relationship demonstrates the expected link between MW and SUVA but not does not mean that the relationship is fixed and one can use it to determine MW in other systems.

Line 168. "The value of S increases with the decrease of the CDOM absorption coefficient". This is not true. It depends on the values of the end members (see Stedmon and Markager 2003). Line 169. Include reference for relationship between S and aromatic content/molecular weight
Line 172. Several of these references did not even measure or report the spectral slope at 275-295.

Line 182. First sentence is repetition.

Line 190. What do you mean by spectral dependency of S275-295? The spectral range should be constant.

Line 193-209. Why not expand the comparison of slope values and ratios with data available from other Siberian rivers Eg. In Walker et al 2013 doi: 10.1002/2013JG002320 (they have seasonal data to compare to). Stedmon et al 2011; Mann et al 2014 & 16. And Gonçalves-Araujo et al 2015 10.3389/fmars.2015.00108

Line 201-214. Is this analysis/interpretation only based on the 2004 data. Why not expand to include all data and compare where you see the qualitative change with where there also is a large drop in CDOM? Is it at the same region the drop in SUVA occurs across all years or is it more salinity that is driving the drop seen in the figure?

Figure 9a and b. It would be more robust to derive the relationship for the 2004 data and test in on the data from other years. I wonder if you carried out the regression analysis between DOC and salinity if you get the same predictive power. The data here look to be very conservative. Mixing is dominating.

---

## Author Comment (AC1) · 8 Sep 2017

Response to review comments on "DOM and its optical characteristics in the Laptev and East Siberian seas: Spatial distribution and inter-annual variability (2003–2011)" by Svetlana P. Pugach et al.

Anonymous Referee #1

General Comments:

In the manuscript entitled "DOM and its optical characteristics in the Laptev and East Siberian seas: Spatial distribution and inter-annual variability (2003-2011)" the authors describe the dominant factors which control the distribution of dissolved organic matter (DOM) in the Siberian shelf seas. On the basis of a data set of several years the authors also try to explain the reason for the observed year-to-year variability of the DOM distribution. A further focus of the work is on the estimation of the utility of in situ fluorescence measurement.

Basically, the article describes phenomena that have been already investigated and published by other authors, i.e.: - The atmospheric forcing of the Lena freshwater plume - and thus the associated high content of terrestrial DOM (for example: Dmitrenko et al., 2005, Wind-driven surface hydrography of the eastern Siberian shelf, doi:10.1029/2005GL023022).-

- The geochemical behavior of DOM on the Laptev and East Siberian Seas (for example: Alling et al., 2010, cited in the manuscript) – The usefulness of in situ fluorescence measurements for the investigation of DOM in the East Siberian and Laptev Seas (Belzile et al., 2004; cited in the manuscript). Actually, some of the co-authors of this manuscript were also co-authors of the study published by Belzile. Because both studies are based on samples from the same region and the same year, the question is whether it is the same set of samples? What is the main difference to the work of Belzile et al.?

*SP: Thank you for your careful consideration of our results. Indeed, N. Shakhova and I. Semiletov co-authored paper by Belzile et al. (2006) which is based on a very limited data set (14 stations, number of samples = 23) obtained on one single cruise in 2004 in a limited nearshore zone of the East Siberian Sea. In contrast, the current paper is based on four much longer expeditions, with orders of magnitude more observations over much greater spatial scales, and using more techniques/parameters. Belzile et al. (2006) show that good estimates of dissolved absorption can be obtained from DOM-FL measured in situ on unfiltered samples, but this work don't study relationship between DOC and CDOM concentrations.*

*The current paper contributes much novel data relative to Belize et al., Dmitrenko et al. (no optical data at all), and Alling et al. (very limited optical data used only as ancillary information). For instance, the utility of high-resolution DOM-FL (> 1750 measurements made only in 2008) and at 286 different stations for biogeochemical studies is assessed by in situ multi-year data obtained in the two very different regimes of the East Siberian Sea, and the Laptev Sea. The spectral characteristics of CDOM and salinity in different East Siberian Arctic Shelf (ESAS) areas (obtained during the 2004, 2005, and 2011 surveys) were used to separate the ESAS into western and eastern biogeochemical provinces (Table 2). Moreover, based on approach described by Weishaar et al., (2003): specific UV absorbance (SUVA) - defined as the UV absorbance of a water sample at 254 nm normalized for DOC concentration, was used to estimate the degree of aromaticity in bulk CDOM. For the first time, CDOM/DOC interannual variability in connection with atmospheric pressure fields and dynamics/wind-driven water circulation is considered in our paper. Taken together, the current paper, relative to earlier studies focusing on hydrology and bulk DOC distribution, focus on DOM dynamics and its optical properties which provides new insight in biogeochemical processes in the ESAS – the broadest and shallowest shelf in the World Ocean.*

In general, it is hard for me to recognize on which data set the analysis is based on. According to Table 1, CDOM (for absorbance measurement) was sampled on 245 stations, but in Table 2 (statistics) the number of samples (N) is 90. Also Figure 2a and 5 give no explanation because the gridded maps do not show the sampling locations for CDOM. Figure 9 (DOC) only shows ~19 sampling locations. This makes it difficult to follow the study's line of argumentation - at least for me. In my opinion, the presentation of the data has to be revised.

*SP: We agree and will revise accordingly. We will give detailed explanation on which data set each analysis is based on. Table 2 will be edited accordingly. The sampling locations for CDOM also will be given in Figures 2a and 5. In the current Figure 9 we did show only 19 sampling locations because at these sites salinity was < 24.5 psu; in areas with higher salinity correlation between CDOM and DOC is going down. Figure 9 will be redrawn showing all sites. Finally, the presentation of the data will be revised.*

The authors distinguish a western biogeochemical province from an eastern province and state that the CDOM concentration is high near the Lena Delta and low in the eastern East Siberian Sea, a region that is less influenced by riverine freshwater. They also found a strong negative correlation between the CDOM absorption and the salinity and concluded that the CDOM is mainly of terrestrial origin. For the Arctic Ocean, this is a well-documented hypothesis (see the work of R. Amon, C. Stedmon; M. Granskog and many others). Unfortunately, no figures were presented to show the correlation between salinity and CDOM. In my opinion, we are not dealing here with different biogeochemical regimes but with differences in the hydrography and spatial expansion of the region of freshwater influence (ROFI) driven by atmospheric forcing. The conclusion that "the atmospheric circulation regime is the dominant factor controlling CDOM spatial distribution" is therefore somewhat misleading. Are the geochemical processes fundamentally different within and outside of the ROFI? This point needs to be discussed in much greater detail.

*SP: Thank you for this comment, which helps to improve our manuscript. In this paper, we don't pretend to be the first who predicted a strong negative correlation between the CDOM absorption and the salinity and concluded that the CDOM is mainly of terrestrial origin. Our objective is to deliver to scientific community our unique multi-year CDOM data sets obtained in the poorly explored shallow ESAS. As requested, a new figure will be presented to show the correlation between salinity and CDOM.*

*Concerning the comment: "we are not dealing here with different biogeochemical regimes but with differences in the hydrography and spatial expansion of the region of freshwater influence (ROFI) driven by atmospheric forcing", we will give our explanation showing that there are no contradiction between reviewer's 1 and our understanding for such a matter. We use an approach described by Semiletov et al., 2005 and Pipko et al., 2005, which shows that based on distribution of the hydrological (and hydrochemical data), two biogeochemical provinces (areas) were identified in the shallow ESAS: a western area (heterotrophic biogeochemical province) that is influenced strongly by freshwater flux and particulate material transport of the coastal eroded material enriched by organic carbon (oxidized to $CO_2$, Semiletov et al., 2016, and other publications cited in this paper), and an eastern area (autotrophic) which is under the influence of highly productive Pacific-derived waters. From year to year, depending from dominated wind-regimes, the longitudinal shift of the boundary (frontal zone) between western and eastern areas may reach 10 degrees and more. In this paper, in addition to hydrological and hydrochemical data, the CDOM and its spectral characteristics were applied to identify different biogeochemical provinces in the surveyed area. We will rewrite our*

*conclusion in this sense as following: "The atmospheric circulation regime is the dominant factor controlling hydrography and spatial expansion of the western area of freshwater influence which determines CDOM/DOM spatial distribution on the ESAS. A western and an eastern regime of ESAS, separated around 165-170° E, were reflected with distinctly different DOM optical properties".*

Specific and technical comments:

- Are the DOM data accessible? - Please indicate the sampling locations in the figures.

*SP: The sampling locations will be shown in the figures. DOM and all data will be made publicly available at the open-access Stockholm University Bolin Centre Database (http://bolin.su.se/data/), once the manuscript is published.*

- I guess you used the DIVA tool (ODV) for the spatial gridding. I believe the interpretation of data can be misleading if a spatial interpolation that is based on 17 sampling locations covers an area of approx. 500.000 square kilometers (Figure 9.). The DIVA interpolation gives DOC and CDOM concentrations ~400 km north of the last sampling location. I would like to suggest to redraw all figures and to show the original data (e.g. as colored dots) instead of the interpolated data.

*SP: Thank you, we agree. To avoid any misinterpretation, the Figure 9 was redrawn as shown below. In the upper row-left, original DOC data are presented as colored dots, in the upper row-right, calculated DOC values are shown as colored dots. In the lower row, the same values, but used ODV for the spatial gridding, are presented. We can find a good visual agreement between interpolated original and calculated DOC distribution.*

[Figure]

*Figure 9. DOC distribution in the ESAS surface waters in September 2004. Upper panels, data presented as colored dots: (a) measured DOC data, (b) DOC data, calculated from DOM-FL. Low panels, interpolated data: (c) measured DOC data, (d) DOC data, calculated from DOM-FL.*

- Samples were taken from a "seawater intake system" that pumped the water into a 300 l barrel. Have you measured the salinity directly in the sample, or did you use the salinity data from the Seabird CTD?

The inlet of the ship was at 4m water depth. If you have taken the salinity data from the CTD, which depth have you chosen?

*SP: We have measured the salinity directly in the barrel using the CTD Seabird 19+ equipped with the WetStar CDOM sensor. "In-barrel" salinity data agreed well with the salinity data measured at stations at depth ~4m. This information will be explicitly added to the revised ms.*

- Please do not use acronyms in the heading of the manuscript

*SP: Agree. Fixed.*

- Page 3 line 147: I guess 0.7 µm is the correct nominal pore size of the filters.

*SP: Thank you. It was our typo. This sentence was rewritten as following: "Water samples for CDOM underwent filtration through 0,7 µm GF/F filters (Whatman, Inc.)".*

- Is figure 3 really necessary? What does it tell about the CDOM distribution?

*SP: We think that for the scientific community and further investigations it would be useful to represent in Figure 3 the surface CDOM absorption coefficient spectra a (λ) obtained for three distinct sites located in contrasting hydrological and biogeochemical conditions.*

- Figure 4: Wouldn't it be better to show the relation between aromaticity and salinity instead of longitude?

*SP: Agree. The relation between aromaticity and salinity is added in a new Figure 4.*

[Figure]

*Figure 4. The relationship between the aromatic carbon content ($C_{Ar}$, %) and salinity in the ESAS surface waters, September 2004.*

- English is not my first language, but I believe the text needs a linguistic revision.

*SP: Agree. The final version of our manuscript will be edited by professional English native editor.*

---

## Author Comment (AC2) · 8 Sep 2017

Response to review comments on "DOM and its optical characteristics in the Laptev and East Siberian seas: Spatial distribution and inter-annual variability (2003–2011)" by Svetlana P. Pugach et al.

Anonymous Referee #2

Review of Svetlana et al DOM and its optical characteristics in the Laptev and East Siberian seas: Spatial distribution and inter-annual variability (2003-2011). The manuscript has the aims of reporting on the inter-annual variability in CDOM and DOC in the Laptev and East Siberian sea. It reads very much like a cruise report and would benefit from a more comprehensive data analysis and discussion of the results obtained.

*SP: Thank you for all your comments, which help us to improve our manuscript further.*

*While we respectfully disagree that our manuscript is "like a cruise report", we recognize there is indeed several aspects that we can improve. A more comprehensive data analysis and more detailed discussion is now facilitated, for instance, by five new figures (shown in the end of this response):*

*- vertical distribution of CDOM along two the west-to-east transects across the ESAS (Figure 1_add);*

*- relationship between DOM fluorescence (DOM-FL) measured using WETStar fluorometer and absorption coefficient at 370 nm ($a_{370}$) in the ESAS (our data) and Belzile data (Figure 2_add);*

*- depth profiles of salinity (a) and DOM-FL (b) measured at two typical stations in the ESAS (their locations are shown in Figure 3), and relations between DOM-FL and salinity (c) (Figure 3_add);*

*- relationship between the DOM-FL and salinity; $S_R$ and salinity in the ESAS surface waters, September 2005 (a) and 2011 (b) (Figure 4_add),*

*- relationship between the surface salinity and DOM-FL (Figure 5_add).*

*This further analysis will be elaborated in the revised discussion section.*

The referencing of previous literature is suboptimal and at times inappropriate. In my opinion there is a missed opportunity for a solid analysis on the linkage between CDOM absorption, fluorescence and DOC across several years.

*SP: The referencing of previous literature was now carefully reconsidered and complemented by several additional references. The linkage between CDOM absorption, fluorescence and DOC was considered in detail only for summer 2004, because only this year all these parameters were investigated simultaneously.*

Why not take more inspiration from the Belzile paper cited and include a comparison of your data with theirs from the East Siberian sea? Does the same FDOM to CDOM relationship exist?

*SP: In the revised ms, we have now compared the relationship between DOM-FL and absorption coefficient at 370 nm ($a_{370}$), which quantify CDOM, using the East Siberian Sea' data reported by Belzile (2006) and our data obtained in the Laptev and East Siberian seas. Figure 2_add (shown below) demonstrates a good agreement between Belzile' and our data. Correlation coefficient between DOM-FL and CDOM – in terms of $a_{370}$, have been found the same in the ESAS (r=0.97, N=92) and Belzile samples (Boothbay Harbor, West Harbor Pond, Beaufort Sea and East Siberian Sea): (r=0.98, N=74).*

*Highest values of DOM-FL and $a_{370}$ can be seen in the Laptev Sea (marked as grey circles) which is strongly impacted by the Lena River runoff.*

The section on the inter-annual variability is difficult for the reader to follow as is. It would likely be easier if figure 5 and 6 were combined so that the sea level pressure maps could be compared with the CDOM and salinity distribution maps. Alternatively, the authors could just compare the maps of both salinity and SLP, then in a separate figure reveal how robust the salinity CDOM relationship was. In this form the manuscript is not suitable for publication and I recommend re-submission after revising the data analysis.

*SP: As espoused above and further below, the ms is significantly revised to further the data analysis and interpretations. On the more specific reviewer comment, we tried to redo our Figures 5 and 6 according to the comment, but we found that of the output was not so useful because of different formats of the datatypes. Instead, we will add a separate figure, which reveals a relationship between the surface salinity and DOM-FL, to the existing figures 5 and 6.*

Whilst doing this you should consider splitting the results and discussion sections to allow for a better separation between your results and reflections on how your findings link to other studies.
*SP: We have revised the data analysis and splitted up the results and discussion sections to allow for a better separation between our results and reflections on how our findings link to other studies.*

Other points to address:

Line 8. "amount" rather than "volume".

*SP: Corrected.*

Line 21. Replace "were" with "was"

Try to avoid use of "e.g." in referencing and citing very many studies. Find the most relevant and limit it to 3-4.

*SP: Agree. Corrected/adjusted throughout the ms.*

Line 46. Replace "gives input" with "supplies".

*SP: Corrected.*

Line 50-51. I suggest you specify this more. Many rivers and streams have high or higher DOC but few large rivers have concentrations this high at their mouth.

*SP: Thank you. This part of introduction was rewritten as:*

*Annually, the Arctic rivers transport 25-36 Tg of DOC to the Arctic Ocean, which is ~10 % of the global riverine DOC discharge (Raymond et al., 2007). The Siberian rivers have high DOC concentration with a mean of more than 500 μM (Gordeev et al., 1996; McClelland et al., 2012; Amon et al., 2012). These concentrations are an order of magnitude higher than in the inflowing Atlantic (60 μM) and Pacific waters (70 μM), but the volume flux of the latter is about 60 times larger than that of continental runoff (Anderson and Amon, 2015). Furthermore, Arctic and subarctic regions contain approximately 50 % of the global terrestrial OC in their frozen soils (Tarnocai et al., 2009; Hugelius*

*et al., 2011). Warming and intensification of the hydrologic cycle is leading to increased rate of water and dissolved organic matter (DOM) discharge from the Siberian rivers (Semiletov et al., 2000; Savelieva et al., 2000; Stein and Macdonald, 2004).*

Line 54. Replace "lead" with "is leading to"

*SP: Corrected.*

Line 56-68. This section should be rephrased and better references found. If you do not want to have too many references I recommend you pick either the original papers or first to demonstrate this in the Arctic. Currently there is a bizarre selection of studies cited and not all directly relevant.

*SP: This section is rephrased and referencing edited.*

Line 78-79. Several of these references are not even Arctic.

*SP: Agree. Corrected.*

Line 80. Were there not any additional scientific aims or hypothesises? Possibly developed during the data analysis for this study? Try to mention them here. As stated now the aim reads very much as a data report.

*SP: Thank you. It was edited to the following:*

*The purpose of this paper is: (1) to study the inter-annual dynamics and optical characteristics of DOM in shelf waters of the Eastern Arctic seas on the basis of multi-year summertime (August – September) expedition data (2003-2005, 2008, 2011); (2) to examine the relationship between CDOM and DOC in order to validate a method for accurate prediction of DOC concentration from CDOM properties; (3) to show the possibility to determine the distribution of terrigenous dissolved organic matter and continental runoff in the surface water of East Siberian Shelf by CDOM optical characteristics.*

Delete line 83-86. This has been established in the Introduction.

*SP: Deleted. Thanks.*

Line 95. Check your phrasing of "would be oxidised".

*SP: Thanks. This sentence was replaced by:*

*Moreover, it has been found that in the past the Lena River played a dominant role in sediment discharge, flushing out soil OM from its vast watershed (Tesi et al., 2016); a significant fraction of "fresh" terrestrial OM contributes to the DOM pool (Karlsson et al., 2016).*

Line 133-134. Delete this. It is a standard fluorometer which is readily available. No need for this. Also the description of the interior optics can be removed. Not really necessary and appears to be copy pasted word for word from Belize et al 2006 paper, which is a little alarming.

*SP: Thanks. It was edited as following:*

***3.3.1 In situ measurements of CDOM fluorescence***

*CDOM fluorescence (DOM-FL) was measured with a WETStar DOM fluorometer which is suitable for in situ measurements without prior filtration of water (Belzile et al., 2006).*
Line 146. What ranges? I do not understand.

*SP: Thank you. It was a typo. This sentence is replaced by: «Water samples for CDOM underwent filtration through 0,7 µm GF/F filters (Whatman, Inc.).*

Lin 150. It is not valid to apply the fit across this range. The spectrum does not behave exponentially and in many samples there will be a shoulder at 280. Additionally the absorption below 240 will be mainly due to other constituents.

Line 156. Sr is not explained, and the whole this part if poorly written.

Line 157. SUVA is not that recent and include a citation of the original paper for this (Weishaar).

Line 158. I do not agree with this sentence. Starting "The last parameter. . ."

Line 161. I do not agree with this extrapolation. The relationship demonstrates the expected link between MW and SUVA but not does not mean that the relationship is fixed and one can use it to determine MW in other systems.

Line 168. "The value of S increases with the decrease of the CDOM absorption coefficient". This is not true. It depends on the values of the end members (see Stedmon and Markager 2003).

Line 169. Include reference for relationship between S and aromatic content/molecular weight

Line 172. Several of these references did not even measure or report the spectral slope at 275-295.

*SP: This section from line 150 was rewritten as following.*

**3.3.2 CDOM optical properties**

*Spectroscopic analysis of CDOM samples was performed using a UNICO 2804 spectrophotometer with a 1 cm quartz cuvette over the spectral range from 200 to 600 nm at 1 nm intervals. Milli-Q (Millipore) water was used as the reference for all samples. Water samples for CDOM underwent filtration through acid-washed Whatman glass fiber filters (GF/F, nominal pore size 0.7 µm).*

*The absorption coefficient ($a\lambda$, m-1) was calculated as follows:*

$$a\,(\lambda) = 2.303A(\lambda)/L, \tag{1}$$

*where A ($\lambda$) is optical density at wavelength $\lambda$, and L is the cell pathlength in meters.*

*The absorption coefficient at 350 nm ($a_{350}$) was chosen to quantify the concentrations of CDOM because of its correlations to DOC and to permit comparison with other results (Spencer et al., 2009; Stedmon et al., 2011; Walker et al., 2013; Gonçalves-Araujo et al., 2015; Mann et al., 2016).*

*The dependence of a ($\lambda$) on $\lambda$ is described using Equation (2):*

$$a\,(\lambda) = a\,(\lambda_0)\,exp\,\{-S\,(\lambda - \lambda_0)\}, \tag{2}$$

*where $a(\lambda_0)$ is the absorption coefficients at reference wavelength $\lambda_0$, and S is a spectral slope defining spectral dependence of the absorption coefficient resulting from CDOM presence (Blough and Del Vecchio, 2002).*

*The spectral slope, S, indicates the rate at which the CDOM absorption coefficient decreases with wavelength increase (Carder et al., 1989). The value of S varies with the source of the CDOM, aromatic content, and molecular weight (Blough and Del Vecchio, 2002; Helms, 2008; Granskog et al., 2012). In near-shore regions, which are under the influence of terrestrial sources with high concentrations of CDOM, S values increases due to the conservative mixing of terrestrial CDOM (high aλ, low S) with oceanic CDOM (low aλ, high S) (Stedmon and Markager, 2003). Therefore, it is also widely accepted that the spectral slope S can be used as a proxy for CDOM composition (Kowalczuk et al., 2003). However, its usefulness is limited by the fact that S depends on the wavelength interval over which it is calculated (Carder et al., 1989; Stedmon et al., 2000). Following recommendations by Helms et al. (2008) a wavelength interval of 275-295 nm was chosen for detailed spectral analysis because it demonstrates the biggest variability of optical parameters under mixing conditions of water with contrasting optical characteristics. The ratio of S values from the shorter (275-295 nm) and the longer wavelength region (350–400 nm), termed the slope ratio, $S_R$, was calculated as described by Helms et al. (2008). $S_R$ values for terrestrial CDOM typically are <1 whereas oceanic CDOM and extensively photodegraded terrestrial CDOM are typically >1.5 (Stedmon and Nelson, 2015).*

*Specific UV absorbance (SUVA) is defined as the UV absorbance of a water sample at 254 nm normalized for dissolved organic carbon (DOC) concentration is used to estimate the degree of aromaticity in bulk CDOM (Weishaar et al., 2003):*

*$C_{Ar} = 6.52*SUVA + 3.63,$*                 (3)

*where $C_{Ar}$ is percentage of aromatic carbon of the total carbon.*

*This equation is applicable for a wide range of aquatic environment(seas, bogs, lakes) since the authors used humic substances that have different chemical characteristics and demonstrated a strong correlation (r = 0.98) between the specific UV absorbance and aromatic carbon content (Weishaar et al., 2003).*

*To calculate concentration of lignin which is a well-established biomarker of terrigenous dissolved organic matter (DOM) in the ocean and has been successfully applied as a tracer of riverine inputs in the Arctic Ocean (Opsahl and Benner, 1997; Opsahl et al., 1999), we used model by Fichot et al., (2016). Exploration of lignin and CDOM relationships provided useful information for the development of two simple empirical models for the retrieval of the sum of nine lignin phenols (TDLP9, nmol $L^{-1}$) from a (λ) in coastal waters (Fichot et al., 2016):*

*when $a_{250} < 4$ m−1, a "low-CDOM" sub-model based on a simple linear regression was used,*

*$ln (TDLP9) = 0.7672 \cdot a_{263} − 0.3987,$*            (4)

*When $a_{250} \geq 4$ m−1, a "high-CDOM" sub-model based on a multiple linear regression was used,*

*$ln (TDLP9) = −2.282 \cdot ln (a_{350} − 8.209 \cdot ln (a_{275}) +11.365 \cdot ln (a_{295}) + 2.909$*      (5)

Line 182. First sentence is repetition.

*SP: Deleted.*

Line 190. What do you mean by spectral dependency of S275-295? The spectral range should be constant.

*SP: Thanks. This sentence was rewritten as following:*

*Figure 3 shows CDOM absorption spectrum and calculated values of $S_{275-295}$ for surface waters of three stations located in typical shelf zones: station 118 is under direct Lena River influence (the Laptev Sea), station 60 is located in the ESS (moderate zone of river and ocean waters mixing), and station 97 in Long Strait.*

Line 193-209. Why not expand the comparison of slope values and ratios with data available from other Siberian rivers Eg. In Walker et al 2013 doi: 10.1002/2013JG002320 (they have seasonal data to compare to). Stedmon et al 2011; Mann et al 2014 & 16. And Gonçalves-Araujo et al 2015 10.3389/fmars.2015.00108

*SP: Thank you. We will expand the comparison of slope values and ratios with data available from other Siberian rivers (citing refs, accordingly).*

Line 201-214. Is this analysis/interpretation only based on the 2004 data. Why not expand to include all data and compare where you see the qualitative change with where there also is a large drop in CDOM? Is it at the same region the drop in SUVA occurs across all years or is it more salinity that is driving the drop seen in the figure?

*SP: Yes, this analysis/interpretation is only based on the 2004 data. We will expand this to 2005, and 2011 (no spectral data are available for 2008) with a special emphasizes at the areas with large drop in CDOM. To answer this question the Figure 4 was redrawn and shown below.*

[Figure]

***Figure 4.*** *The relationship between the aromatic carbon content ($C_{Ar}$, %) and salinity, $S_R$ and salinity in the ESAS surface waters, September 2004.*

*As can be seen from new Figure 4, SUVA ($C_{Ar}$= 6.52*SUVA + 3.63) is strongly correlated with salinity (r= -0.7). SUVA and CDOM are also closely related (r = 0.71). Then we can say that large drop in CDOM, Sr and SUVA values is driven by increased salinity across all years; position of the region of large change of CDOM and other parameters demonstrated inter-annual dynamics.*

Figure 9a and b. It would be more robust to derive the relationship for the 2004 data and test in on the data from other years. I wonder if you carried out the regression analysis between DOC and salinity if you get the same predictive power. The data here look to be very conservative. Mixing is dominating.

*SP: Thanks. It seemed that this comment is targeted on Figure 8a and 8b. Yes, mixing is dominating. Following your comment we joint all the data for 2004 and 2008 in new Fig. 8c.*

[Figure]

***Figure 8.*** *DOC concentration (µmol l$^{-1}$) versus DOM fluorescence measured using the WETStar fluorometer (DOM-FL, QSU) in the ESAS surface water, September 2004 (**a**), 2008 (**b**) and combined for two years (**c**); **W** - annual Lena River discharge.*

*As can be seen the DOM-FL-DOC predictive power is high and can likely be used in other studies. Unfortunately, the relationship between DOC and salinity is not so strong, because of melt water that lowers this correlation. For example, in 2004 the pair salinity-DOC demonstrates high negative correlation (0.9) in the ESS nearshore zone, while this correlation drops down to - 0.7 in 2011 in the LS and ESS, because of more meltwater. Thus, the relationship between salinity and DOC can be used only for river-freshened areas (limited by isohaline 24.5) and with caution.*

**Additional Figures**

[Figure]

**a)**

[Figure]

**b)**

***Figure 1_add***. *Vertical distribution of DOM-FL (left panel) and salinity (right panel) along the Lena River transect (a) and the west-to-east transect across the ESAS (b) in September, 2004.*

[Figure]

**Figure 2_add.** *Relationship between DOM fluorescence (DOM-FL) measured using WETStar fluorometer and absorption coefficient at 370 nm ($a_{370}$). Grey – our data in the LS (N = 23), Green – our data in the ESS (N = 69) and red – data from Belzile et al., 2006 (N = 23).*

[Figure]

**Figure 3_add.** *Depth profiles of (A) salinity and (B) DOM-FL measured at two stations on the ESAS, and (C) relations between DOM-FL and salinity.*

[Figure]

***Figure 4_add.*** *The relationship between the DOM-FL and salinity; $S_R$ and salinity in the ESAS surface waters, September 2005 (a) and 2011 (b).*

[Figure]

***Figure 5_add.*** *The relationship between the salinity and DOM-FL in the ESAS surface water in August-September 2003-2005, 2008 and 2011(blue circles – salinity < 24.5, red - > 24.5).*

---

## Author Response (AR1)

The handling editor, Ocean Sciences

Dear Editor:

We appreciate the overall positive appraisal and the detailed and insightful comments of the two reviewers on our manuscript "*DOM and its optical characteristics in the Laptev and East Siberian seas: Spatial distribution and inter-annual variability (2003-2011)*" by Svetlana P. Pugach et al." (manuscript ID OS-2017-20). Their constructive reviews and suggestions have contributed to further improve the paper during our revisions. All referee comments and our responses, as well as the resulting edits, are detailed below including a point-by-point response to the reviews, a list of all relevant changes made in the manuscript, and a marked-up manuscript version.

Our first intention was to separate "Results" and "Discussion". However, in the final stage writing this paper we found that is more effective to join that, because of complexity of achieved results: from the role of hydrometeorological situation to the spatial-time variability of the DOC/CDOM and its optical characteristics. In the revised manuscript, the item "Results and Discussion" was significantly modified. It contains now the following sub-items:

4.1  Hydrometeorological situation and CDOM spatial variability

4.2  Spatial variability of CDOM spectral characteristics

4.3  Optical characteristics of the ESAS biogeochemical provinces

4.4  Rapid assessment of dissolved organic carbon based on optical characteristics of dissolved organic matter.

A more comprehensive data analysis and more detailed discussion is now fortified, for instance, by five new figures and associated discussion, a few figures were modified, the fragment related with a goal and objectives, as well as conclusions was rewritten accordingly.

We very hope that the final version of our manuscript fits well to the high standards of the *Ocean Sciences.*

Thank you very much for your consideration,

Svetlana Pugach,
on behalf of all co-authors

**Author responses to reviews, and edits to manuscript number OS-2017-20 "*DOM and its optical characteristics in the Laptev and East Siberian seas: Spatial distribution and inter-annual variability (2003-2011)*" by Svetlana P. Pugach, Irina I. Pipko, Natalia E. Shakhova, Evgeny A. Shirshin, Irina V. Perminova, Örjan Gustafsson, Valery G. Bondur, Alexey S. Ruban, and Igor P. Semiletov.**

**Anonymous Referee #1**

General Comments:

In the manuscript entitled "DOM and its optical characteristics in the Laptev and East Siberian seas: Spatial distribution and inter-annual variability (2003-2011)" the authors describe the dominant factors which control the distribution of dissolved organic matter (DOM) in the Siberian shelf seas. On the basis of a data set of several years the authors also try to explain the reason for the observed year-to-year variability of the DOM distribution. A further focus of the work is on the estimation of the utility of in situ fluorescence measurement.

Basically, the article describes phenomena that have been already investigated and published by other authors, i.e.: - The atmospheric forcing of the Lena freshwater plume - and thus the associated high content of terrestrial DOM (for example: Dmitrenko et al., 2005, Wind-driven surface hydrography of the eastern Siberian shelf, doi:10.1029/2005GL023022).- The geochemical behavior of DOM on the Laptev and East Siberian Seas (for example: Alling et al., 2010, cited in the manuscript) - The usefulness of in situ fluorescence measurements for the investigation of DOM in the East Siberian and Laptev Seas (Belzile et al., 2004; cited in the manuscript). Actually, some of the co-authors of this manuscript were also co-authors of the study published by Belzile. Because both studies are based on samples from the same region and the same year, the question is whether it is the same set of samples? What is the main difference to the work of Belzile et al.?

*Thank you for your careful consideration of our results. Indeed, N. Shakhova and I. Semiletov co-authored paper by Belzile et al. (2006) which is based on a very limited data set (14 stations, number of samples = 23) obtained on one single cruise in 2004 in a limited nearshore zone of only the East Siberian Sea (not Laptev Sea). In contrast, the current paper is based on four much longer expeditions, with orders of magnitude more observations over much greater spatial scales, and using more techniques/parameters. The results of the early/pioneering but limited study of Belzile et al. (2006) indeed gave some indications that estimates of dissolved absorption may be obtained from CDOM measured in situ on unfiltered samples, but that work did not at all study the relationship between DOC and CDOM concentrations.*

*The current paper contributes much novel data and interpretations/insights relative to Belzile et al., 2006; Dmitrenko et al., 2005 (**no optical data at all**), and Alling et al., 2010 (**very limited optical data used only as ancillary information**). For instance, the utility of high-resolution CDOM (> 1750 measurements made only in 2008) and at 286 different stations for biogeochemical studies is assessed by in situ multi-year data obtained in the two very different regimes of the East Siberian Sea, and the Laptev Sea. The spectral characteristics of CDOM and salinity in different East Siberian Arctic Shelf (ESAS) areas (obtained during the 2004, 2005, and 2011 surveys) were used to separate the ESAS into western and eastern biogeochemical provinces (Table 2). Moreover, based on the approach described by Weishaar et al., (2003), specific UV absorbance (SUVA) - defined as the UV absorbance of a water sample at 254 nm normalized for DOC concentration, was used to estimate the degree of aromaticity in bulk CDOM. For the first time, CDOM/DOC interannual variability in connection with atmospheric pressure fields and dynamics/wind-driven water circulation is considered in our paper. Taken together, the current paper, relative to earlier studies focusing on hydrology and bulk DOC distribution, focuses on DOM dynamics and its optical properties and thereby provides new insight in biogeochemical processes in the ESAS - the broadest and shallowest shelf in the World Ocean.*

In general, it is hard for me to recognize on which data set the analysis is based on. According to Table 1, CDOM (for absorbance measurement) was sampled on 245 stations, but in Table 2 (statistics)

the number of samples (N) is 90. Also Figure 2a and 5 give no explanation because the gridded maps do not show the sampling locations for CDOM. Figure 9 (DOC) only shows ~19 sampling locations. This makes it difficult to follow the study's line of argumentation - at least for me. In my opinion, the presentation of the data has to be revised.

*We agree that clarity on these aspects may be improved. The number of samples used in the Table 2 was determined by the number of spectral characteristic measurements (Table 1). The sampling locations for CDOM were given in Figure 2b. In the current Figure 10a we showed only 19 sampling locations because at these sites salinity was < 24.5 psu (the total number of the DOC measurements is 20); it is explained in ms that in areas with higher salinity, the correlation between CDOM and DOC is breaking down.*

The authors distinguish a western biogeochemical province from an eastern province and state that the CDOM concentration is high near the Lena Delta and low in the eastern East Siberian Sea, a region that is less influenced by riverine freshwater. They also found a strong negative correlation between the CDOM absorption and the salinity and concluded that the CDOM is mainly of terrestrial origin. For the Arctic Ocean, this is a well-documented hypothesis (see the work of R. Amon, C. Stedmon; M. Granskog and many others). Unfortunately, no figures were presented to show the correlation between salinity and CDOM. In my opinion, we are not dealing here with different biogeochemical regimes but with differences in the hydrography and spatial expansion of the region of freshwater influence (ROFI) driven by atmospheric forcing. The conclusion that "the atmospheric circulation regime is the dominant factor controlling CDOM spatial distribution" is therefore somewhat misleading. Are the geochemical processes fundamentally different within and outside of the ROFI? This point needs to be discussed in much greater detail.

*Thank you for this comment, which helps to improve our manuscript. In this paper, we don't pretend to be the first who predicted a strong negative correlation between the CDOM absorption and the salinity and concluded that the CDOM is mainly of terrestrial origin. Our objective is to deliver to scientific community our unique multi-year CDOM data sets obtained in the poorly explored shallow ESAS. As requested, a new (Figure 3) was presented to show the correlation between salinity and CDOM.*

*Concerning the comment: "we are not dealing here with different biogeochemical regimes but with differences in the hydrography and spatial expansion of the region of freshwater influence (ROFI) driven by atmospheric forcing", we give our explanation showing that there are no contradiction between reviewer's 1 and our understanding for such a matter. We use an approach described by Semiletov et al., 2005 and Pipko et al., 2005, which shows that based on distribution of the hydrological (and hydrochemical data), two biogeochemical provinces (areas) were identified in the shallow ESAS: a western area (heterotrophic biogeochemical province) that is influenced strongly by freshwater flux and particulate material transport of the coastal eroded material enriched by organic carbon, and an eastern area (autotrophic) which is under the influence of highly productive Pacific-derived waters. From year to year, depending from dominated wind-regimes, the longitudinal shift of the boundary between western and eastern areas may reach 10 degrees and more. In this paper, in addition to hydrological and hydrochemical data, the CDOM and its spectral characteristics were applied to identify different biogeochemical provinces in the surveyed area. (New section 4.3).*

Specific and technical comments:

- Are the DOM data accessible? - Please indicate the sampling locations in the figures.

*The sampling locations were shown in Figure 2. DOM and all data will be made publicly available at the open-access Stockholm University Bolin Centre Database ([http://bolin.su.se/data/](http://bolin.su.se/data/)), once the manuscript is published.*

- I guess you used the DIVA tool (ODV) for the spatial gridding. I believe the interpretation of data can be misleading if a spatial interpolation that is based on 17 sampling locations covers an area of approx. 500.000 square kilometers (Figure 9.). The DIVA interpolation gives DOC and CDOM

concentrations ~400 km north of the last sampling location. I would like to suggest to redraw all figures and to show the original data (e.g. as colored dots) instead of the interpolated data.

*To avoid any misinterpretation, the Figure 9 was redrawn as shown below. In the upper row-left, original DOC data are presented as colored dots, in the upper row-right, calculated DOC values are shown as colored dots. In the lower row, the same values, but used ODV for the spatial gridding, are presented. We can find a good visual agreement between interpolated original and calculated DOC distribution. So, we slightly changed the Figure 9 (Figure 10 in the revised ms).*

[Figure]

*Figure 9. DOC distribution in the ESAS surface waters in September 2004. Upper panels, data presented as colored dots: (a) measured DOC data, (b) DOC data, calculated from CDOM. Low panels, interpolated data: (c) measured DOC data, (d) DOC data, calculated from CDOM.*

- Samples were taken from a "seawater intake system" that pumped the water into a 300 l barrel. Have you measured the salinity directly in the sample, or did you use the salinity data from the Seabird CTD?

The inlet of the ship was at 4m water depth. If you have taken the salinity data from the CTD, which depth have you chosen?

*We have measured the salinity directly in the barrel using a secondary CTD Seabird 19+ equipped with the WetStar CDOM sensor and fully immersed in the barrel. "In-barrel" salinity data agreed well with the salinity data measured at stations at depth ~4m using the regular NISKIN Rosette CTD. This information was explicitly added to the revised ms (Lines 120-125).*

- Please do not use acronyms in the heading of the manuscript

*Agree. Fixed.*

- Page 3 line 147: I guess 0.7 pm is the correct nominal pore size of the filters.

*Thank you. It was our typo. This sentence was rewritten as following: "Water samples underwent filtration through acid-washed Whatman glass fiber filters (GF/F, nominal pore size 0.7 μm)" (Lines 138-139).*

- Is figure 3 really necessary? What does it tell about the CDOM distribution?

*We think that for the scientific community and further investigations it would be useful to represent in Figure 3 (Figure 5 in the revised ms) the surface CDOM absorption coefficient spectra $a_\lambda$ obtained for three distinct sites located in contrasting hydrological and biogeochemical conditions.*

- Figure 4: Wouldn't it be better to show the relation between aromaticity and salinity instead of longitude?

*The relation between aromaticity and salinity is now presented in a new Figure 6.*

- English is not my first language, but I believe the text needs a linguistic revision.

*The final version of our manuscript was edited by a professional English native editor.*

**Anonymous Referee #2**

Review of Svetlana et al DOM and its optical characteristics in the Laptev and East Siberian seas: Spatial distribution and inter-annual variability (2003-2011). The manuscript has the aims of reporting on the inter-annual variability in CDOM and DOC in the Laptev and East Siberian sea. It reads very much like a cruise report and would benefit from a more comprehensive data analysis and discussion of the results obtained.

*Thank you for all your comments, which helped us to improve our manuscript further.*

*While we respectfully disagree that our manuscript is "like a cruise report", we recognize there is indeed several aspects that we can improve. A more comprehensive data analysis and more detailed discussion is now fortified, for instance, by five new figures and associated discussion:*

*- **Figure 3**. The relationship between the salinity and CDOM (QSU) in the ESAS surface water in August-September 2003, 2004, 2005, 2008, and 2011 (blue circles – salinity < 24.5, red - > 24.5);*

*- **Figure 4**. Depth profiles of salinity (**a**) and CDOM (**b**) measured at two stations on the ESAS, and (**c**) relationship between CDOM and salinity;*

*- **Figure 6**. The relationship between CDOM absorption coefficient at the 350 nm wavelength (a$_{350}$, m$^{-1}$), S$_R$ and the aromatic carbon content (C$_{Ar}$, %) and salinity in the ESAS surface waters, September 2004 (**a**), 2005 (**b**), and 2011 (**c**);*

*- **Figure 7**. The relationship between the dissolved lignin concentrations (TDLP$_9$, nmol L$^{-1}$) and salinity in the ESAS surface waters, September 2004 (**a**), 2005 (**b**), and 2011 (**c**), where r$_W$, r$_E$ represent correlation coefficients between TDLP$_9$ and salinity for western and eastern parts of the ESAS, respectively, for these three years combined;*

*- **Figure 8**. Correlation between CDOM (QSU) and absorption coefficient at 350 nm (a$_{350}$, m$^{-1}$) in ESAS surface water (**a**) and relationship between CDOM (QSU) and absorption coefficient at 370 nm (a$_{370}$, m$^{-1}$) (**b**), September 2004. Grey – our data in the Laptev Sea (N = 23), green – our data in the East Siberian Sea (N = 69), and red – data from Belzile et al., 2006 (N = 23).*

*Additionally, a few figures were modified (Figures 1, 2, 9 in the revised ms). The structure of the revised manuscript was significantly modified.*

The referencing of previous literature is suboptimal and at times inappropriate. In my opinion there is a missed opportunity for a solid analysis on the linkage between CDOM absorption, fluorescence and DOC across several years.

*The referencing of previous literature has now been carefully reconsidered and complemented by several additional references.*

*The linkage between CDOM absorption, fluorescence and DOC was considered in detail only for summer 2004, because only this year all these parameters were investigated simultaneously.*

Why not take more inspiration from the Belzile paper cited and include a comparison of your data with theirs from the East Siberian sea? Does the same FDOM to CDOM relationship exist?

*In the revised ms, we have now compared the relationship between FDOM and absorption coefficient at 370 nm (a$_{370}$), which quantify CDOM, using the 2004 East Siberian Sea' data reported by Belzile et al. (2006) and our data obtained in both the Laptev and East Siberian seas. Figure 8 demonstrates a good agreement between Belzile's and our data. Correlation coefficient between FDOM and CDOM - in terms of a$_{370}$, have been found the same in the ESAS (r=0.97, N=92) and Belzile's samples (Boothbay Harbor, West Harbor Pond, Beaufort Sea and East Siberian Sea): (r=0.98, N=74).*

*Highest values of FDOM and a$_{370}$ can be seen in the Laptev Sea (marked as grey circles) which is strongly impacted by the Lena River runoff.*

The section on the inter-annual variability is difficult for the reader to follow as is. It would likely

be easier if figure 5 and 6 were combined so that the sea level pressure maps could be compared with the CDOM and salinity distribution maps. Alternatively, the authors could just compare the maps of both salinity and SLP, then in a separate figure reveal how robust the salinity CDOM relationship was. In this form the manuscript is not suitable for publication and I recommend re-submission after revising the data analysis.

*As espoused above and further below, the ms is significantly revised to further the data analysis and interpretations. We combined Figures 5 and 6 according to the comment (Figure 2 in the revised ms).*

Whilst doing this you should consider splitting the results and discussion sections to allow for a better separation between your results and reflections on how your findings link to other studies.

*Our first intention was to separate "Results" and "Discussion". However, in the final stage writing this paper we found that is more effective to join that, because of complexity of achieved results: from the role of hydrometeorological situation to the spatial-time variability of the DOC/CDOM and its optical characteristics.*

Other points to address:

Line 8. "amount" rather than "volume".

*Corrected.*

Line 21. Replace "were" with "was"

Try to avoid use of "e.g." in referencing and citing very many studies. Find the most relevant and limit it to 3-4.

*Agree. Corrected/adjusted throughout the ms.*

Line 46. Replace "gives input" with "supplies".

*Corrected.*

Line 50-51. I suggest you specify this more. Many rivers and streams have high or higher DOC but few large rivers have concentrations this high at their mouth.

*This part of the Introduction was rewritten as:*

*"Annually, the Arctic rivers transport 25-36 Tg of DOC to the Arctic Ocean, which is ~10 % of the global riverine DOC discharge (Raymond et al., 2007). Siberian rivers are high in DOC, with a mean concentration of more than 500 µM (Gordeev et al., 1996; McClelland et al., 2012; Amon et al., 2012). These concentrations are an order of magnitude higher than in the inflowing Atlantic (60 µM) and Pacific (70 µM) waters, but the volume flux of the two oceans is about 60 times larger than that of continental runoff (Anderson and Amon, 2015). Furthermore, Arctic and subarctic regions contain approximately 50 % of the global terrestrial OC in their frozen soils (Hugelius et al., 2011). Warming and intensification of the hydrologic cycle is leading to an increased rate of water and dissolved organic matter (DOM) discharge from the Siberian rivers (Savelieva et al., 2000; Semiletov et al., 2000; Stein and Macdonald, 2004)." (Lines 62-69).*

Line 54. Replace "lead" with "is leading to"

*Corrected.*

Line 56-68. This section should be rephrased and better references found. If you do not want to have too many references I recommend you pick either the original papers or first to demonstrate this in the Arctic. Currently there is a bizarre selection of studies cited and not all directly relevant.

*This section was deleted.*

Line 78-79. Several of these references are not even Arctic.

*This part was rewritten as:*

*"Empirical relationships between the optical properties of DOM and DOC have already been the subject of investigation in the Arctic Ocean (Stedmon et al., 2011; Guéguen et al., 2005, 2007; Fichot and Benner, 2011; Gonçalves-Araujo et al., 2015; Kaiser et al., 2017). The present study synthesizes the authors' multi-year observations on the remote ESAS with the focus on exploring the extent and dynamics of riverine DOC, using CDOM as a proxy." (Lines 83-86).*

Line 80. Were there not any additional scientific aims or hypothesises? Possibly developed during the data analysis for this study? Try to mention them here. As stated now the aim reads very much as a data report.

*It was edited to the following:*

*"This paper aims (1) to study the spatial and inter-annual dynamics of DOM optical characteristics in shelf waters of the Eastern Arctic seas on the basis of multi-year summertime (August –September) expedition data (2003, 2004, 2005, 2008, 2011); (2) to examine the relationship between CDOM fluorescence and DOC in order to validate a useful method for accurately predicting DOC concentration from CDOM properties in the ESAS; and (3) to demonstrate the feasibility, using DOM optical characteristics, of determining the terrigenous DOM distribution and identifying different biogeochemical provinces in the shelf water." (Lines 87-92).*

Delete line 83-86. This has been established in the Introduction.

*We wanted to highlight these two seas of the ESAS because here the influence of river discharge and the biogeochemical signal of permafrost degradation are the most prominent. The introduction referred to the entire ESAS.*

Line 95. Check your phrasing of "would be oxidised".

*This sentence was replaced by:*

*"Moreover, it has been found that in the past the Lena River played a dominant role in sediment discharge, flushing out soil OM from its vast watershed (Tesi et al., 2016); a significant fraction of "fresh" terrestrial OM contributes to the DOM pool (Vonk et al., 2013; Karlsson et al., 2016)." (Lines 103-105).*

Line 133-134. Delete this. It is a standard fluorometer which is readily available. No need for this.

Also the description of the interior optics can be removed. Not really necessary and appears to be copy pasted word for word from Belize et al 2006 paper, which is a little alarming.

*It was edited as following:*

*"CDOM fluorescence was measured with a WETStar DOM fluorometer, which is suitable for in situ measurements without prior filtration of water (Belzile et al., 2006). The raw voltage from the fluorometer was converted to quinine sulfate units (QSU) (Belzile et al., 2006)." (Lines 176-178).*

Line 146. What ranges? I do not understand.

*It was a typo. This sentence is replaced by: "Water samples underwent filtration through acid-washed Whatman glass fiber filters (GF/F, nominal pore size 0.7 $\mu$m)" (Lines 138-139).*

Line 150. It is not valid to apply the fit across this range. The spectrum does not behave exponentially and in many samples there will be a shoulder at 280. Additionally the absorption below 240 will be mainly due to other constituents.

*This section was rewritten (Lines 136-178 in the revised ms).*

Line 156. Sr is not explained, and the whole this part if poorly written.

*We explained Sr and rewrote this part of the ms (Lines 136-178)*

Line 157. SUVA is not that recent and include a citation of the original paper for this (Weishaar).

*Fixed (Lines 162-165).*

Line 158. I do not agree with this sentence. Starting "The last parameter. . ."

*The sentence was deleted.*

Line 161. I do not agree with this extrapolation. The relationship demonstrates the expected link between MW and SUVA but not does not mean that the relationship is fixed and one can use it to determine MW in other systems.

*This text was clarified (Line 168 in the revised ms).*

Line 168. "The value of S increases with the decrease of the CDOM absorption coefficient". This is not true. It depends on the values of the end members (see Stedmon and Markager 2003).

*This sentence was rewritten as following:*

*"The value of S varies with the source, aromatic content, and molecular weight of the CDOM (Blough and Del Vecchio, 2002; Helms et al., 2008; Granskog et al., 2012). In near-shore regions, which are under the influence of terrestrial sources with high concentrations of CDOM, S values increase due to the conservative mixing of terrestrial CDOM (high $a_\lambda$, low S) with oceanic CDOM (low $a_\lambda$, high S) (Stedmon and Markager, 2003)." (Lines 151-155).*

Line 169. Include reference for relationship between S and aromatic content/molecular weight

*Done. (Line 152)*

Line 172. Several of these references did not even measure or report the spectral slope at 275-295.

*This section was rewritten as following:*

*"Following recommendations by Helms et al. (2008) a wavelength interval of 275-295 nm was chosen for detailed spectral analysis because it demonstrates the biggest variability of optical parameters when mixing waters with contrasting optical characteristics." (Lines 156-159).*

Line 182. First sentence is repetition.

*This sentence was rewritten (Lines 276-277).*

Line 190. What do you mean by spectral dependency of S275-295? The spectral range should be constant.

*This sentence was rewritten as following:*

*"CDOM absorption spectra for September 2004 surface waters at two stations (stations 118 and 97) located in contrasting shelf zones (Fig. 1) are shown at Figure 5. Spectrum for station 60 located in the East Siberian Sea (in a moderate zone of mixing river and ocean waters) is also presented. CDOM absorption spectra measured for different waters have substantially different levels of absorption." (Lines 278-281).*

Line 193-209. Why not expand the comparison of slope values and ratios with data available from other Siberian rivers Eg. In Walker et al 2013 doi: 10.1002/2013JG002320 (they have seasonal data to compare to). Stedmon et al 2011; Mann et al 2014 & 16. And Gon9alves-Araujo et al 2015 10.3389/fmars.2015.00108

*We compared our data with data available from other Siberian rivers in Sections 4.2 and 4.3 of the revised ms.*

Line 201-214. Is this analysis/interpretation only based on the 2004 data. Why not expand to include all data and compare where you see the qualitative change with where there also is a large drop in CDOM? Is it at the same region the drop in SUVA occurs across all years or is it more salinity that is driving the drop seen in the figure?

*Yes, this analysis/interpretation is only based on the 2004 data. We expanded this to 2005, and 2011 (no spectral data are available for 2008) with a special emphasizes at the areas with large drop in CDOM ($a_{350}$) (Figures 6 and 7 in the revised ms).*

*As can be seen from this new Figure 6, SUVA ($C_{Ar} = 6.52*SUVA + 3.63$) is correlated with salinity. SUVA and CDOM ($a_{350}$) are also closely related. Then we can say that the distribution of CDOM ($a_{350}$), Sr and SUVA values is driven by increased salinity across all years. Furthermore, the position of the region of large change of CDOM and other parameters demonstrated inter-annual dynamics.*

Figure 9a and b. It would be more robust to derive the relationship for the 2004 data and test in on the data from other years. I wonder if you carried out the regression analysis between DOC and salinity if you get the same predictive power. The data here look to be very conservative. Mixing is dominating.

*Yes, mixing is dominating. Following this reviewer's comment, we combined the data for 2004 and 2008 (Figure 9 in the revised ms).*

*As seen in Fig. 9, the CDOM-DOC predictive power is high and can likely be used in other studies ($r = 0.99$ for 2004 and 2008). Unfortunately, the relationship between DOC and salinity is not so strong, because of melt water that lowers this correlation. For example, in 2004 the pair salinity-DOC demonstrates high negative correlation (- 0.91) in the ESS nearshore zone, while this correlation drops down to - 0.77 in 2008.*

[revised manuscript text omitted]

¶
<объект><объект>